# A Comparative Analysis for Defining the Sliding Surface and Internal Structure in an Active Landslide Using the HVSR Passive Geophysical Technique in Pujilí (Cotopaxi), Ecuador

Olegario Alonso-Pandavenes [1,*], Daniela Bernal [1], Francisco Javier Torrijo [2] and Julio Garzón-Roca [3]

1. Geology and Mining Engineering Faculty-FIGEMPA, Central University of Ecuador, Quito 170521, Ecuador; cdbernal@uce.edu.ec
2. Research Centre for Architecture, Heritage and Management for Sustainable Development (PEGASO), Department of Geotechnical Engineering, Universitat Politècnica de València, Camino de Vera s/n, 46022 Valencia, Spain; fratorec@trr.upv.es
3. Department of Geodynamics (GEODESPAL), Faculty of Geology, Complutense University of Madrid, 28040 Madrid, Spain; julgarzo@ucm.es
* Correspondence: omalonso@uce.edu.ec; Tel.: +593-995608066

**Abstract:** Geophysical techniques were employed to analyze one of the landslides that affected the main access road to Pujilí (Ecuador). A passive seismic technique was utilized to test a total of 70 horizontal to vertical spectral ratio (HVSR) points, complemented by an active seismic-refraction profile and a multichannel analysis of surface waves (MASW) survey. The results from the active geophysical surveys facilitated the determination of the shear-wave-velocity value for the surface materials that were in motion. However, the HVSR provided the fundamental frequency $f_o$ and amplification $A_o$ values of the ground. The Nakamura (1989) relationship was applied to obtain the thickness of the sediments over a compact material from the fundamental frequency of the terrain in a two-layer model. Additionally, constrained models of the shear-wave velocity ($V_s$) distribution in the landslide area, obtained from the active seismic surveys, were used to invert the ellipticity curves. The results from this inversion were compared with those obtained by applying the Nakamura equation. The landslide-rupture surface was delineated for each type of analysis, which verified the correlation and minimal differences between the results of the three proposed studies, thus validating the procedure. The directivity of the microtremor HVSR signals was also analyzed, demonstrating a relationship with the internal structure of the sliding material. Furthermore, the ability to slide concerning the $K_g$ parameter (vulnerability index, Nakamura, 1989) was studied. The usefulness of the directivity analysis in defining the internal structures in landslide materials and in determining the areas with the most significant instability was demonstrated. Overall, the HVSR is considered valuable when conducting early landslide studies and is helpful in determining the rupture plane while remaining a simple, fast, and economical technique.

**Keywords:** Pujilí landslide; HVSR; MASW technique; surface of rupture; natural frequency $f_o$



## 1. Introduction

Landslides are natural geohazards that have recently caused significant losses, affecting both property and human lives. The largest landslides are recorded in the EMDAT database [1]. Significant sliding events exceeding 100 have been documented, exceeding 100, with more than 15,000 lives lost and over 4 million people affected worldwide. It is essential to consider and expend time and resources to study and prevent them. Depending on the nature of the ground, landslides can develop in soils, rocks, and fillings (natural or artificial). Each geological material has a different shape and rupture type [2,3], and the movement speed can vary, ranging from very slow or creeping to fast and sudden. One of the primary goals of the study of landslide-mitigation actions is to define the rupture or

failure surface [3]. This can be an easy task when the mobilizing materials differ significantly from the static materials, such as when soft sediments slide over a rocky substrate. Otherwise, the differentiation and investigation of the sliding surface when it is inside a homogeneous stratum (i.e., involving identical materials in the mobilization of sliding mass and the one kept fixed) can be very complicated [2,4].

Traditional investigations consider the performance of drill holes and using instrumentation inside (like piezometers and inclinometers). These tests help define the sliding surface and provide information with which to make decisions about its mitigation [5]. However, this type of landslide study is usually expensive and complicated. This is mostly due to the difficulty in accessing the working areas where these events occur, which are often high slopes, and the instabilities in the subsoil [4–6]. The application of geophysical techniques is one of the alternatives to early investigations. These methods can provide broad information and can be used to delimitate the most favorable areas to subsequently investigate in depth [7–9].

Geophysical landslide research has traditionally focused on active techniques, such as using seismic and geoelectrical methods. In recent decades, active seismic refraction and electrical tomography have been used to study and define sliding masses. These investigations obtained excellent results, especially when both techniques were combined to detect the failure surface, alteration levels, water content, and water table, all of which affect landslide phenomena [7,8]. Over the last twenty-five years, the application of passive investigation techniques using the seismic method has become more common, particularly in landslides related to the study of seismic risk and their triggers (co-seismic landslides) [8–10]. In this context, the horizontal to vertical spectral ratio (HVSR) technique has emerged to measure the ground-vibration period in seismic-hazard studies. It is generally accepted that the passive seismic single-station technique determines the natural frequency vibration ($f_o$) on a simple two-layer model, but more advanced applications are still under discussion [11–13].

As in any application of a geophysical method, the importance of the net separation between anomalies and materials is fundamental. In this case, the HVSR technique is based on the ability to separate two materials whose sonic or elastic impedances demonstrate clear differences. This was previously investigated in [14]. Based on those studies, Nakamura [15,16] concluded with the definition of a two-layer model: loose sediments over a competent substrate related to the natural frequency $f_o$ of the area. The HVSR technique has been applied in a range of investigations, including the detection of a rocky substrate under ice, sand, or sediments [17–19], liquefaction-potential estimation [20], and, under constrained conditions, the establishment of the shear-wave velocity ($V_s$) distribution profile through the analysis of the ellipticity curve [21–23]. The HVSR-based passive seismic technique has also been used to study sliding masses, such as cliff areas and landslides, as well as materials' mobilization by gravity [24–31]. Thus, the HVSR survey has the potential to become a passive, non-invasive, and low-cost method for the long or short-term study, monitoring, and characterization of landslides [18,22,23].

In Ecuador, landslide processes typically occur due to dynamic demands on the ground, such as earthquakes, or long periods of rain, some of which are related to El Niño events. They affect various types of geological formation and a vast territory from the coast to the Andean Cordillera. Most of the locations where they occur have significant slopes with loose soils, which complicates the installation of drilling machines [32]. In Pujilí canton (Cotopaxi province, Ecuador), landslides of different magnitudes and velocities of movement have occurred, particularly in the Cachi Alto community and its surroundings. This area is tectonically active and has experienced three medium-to-large-magnitude landslides over several years across a broad region [33,34]. At the beginning of 2018, an area of about 19,000 m$^2$ began to slide, initially affecting only farmlands. The landslide area continued to expand throughout that year and 2019, and it involved a house and the communication route with the Cachi Bajo neighborhood (Figure 1). According to early studies in the area [33–36], the landslide was due to a zone of poorly consolidated material

that slid over a more compact material. In this case, the geological ground corresponds to the same type of material, called *cangahua*, a hardened eruptive volcanic soil that is mostly cemented. The moving mass (altered *cangahua*) and the fixed mass (or cemented material) have very similar geological and geotechnical parameters [35,36].

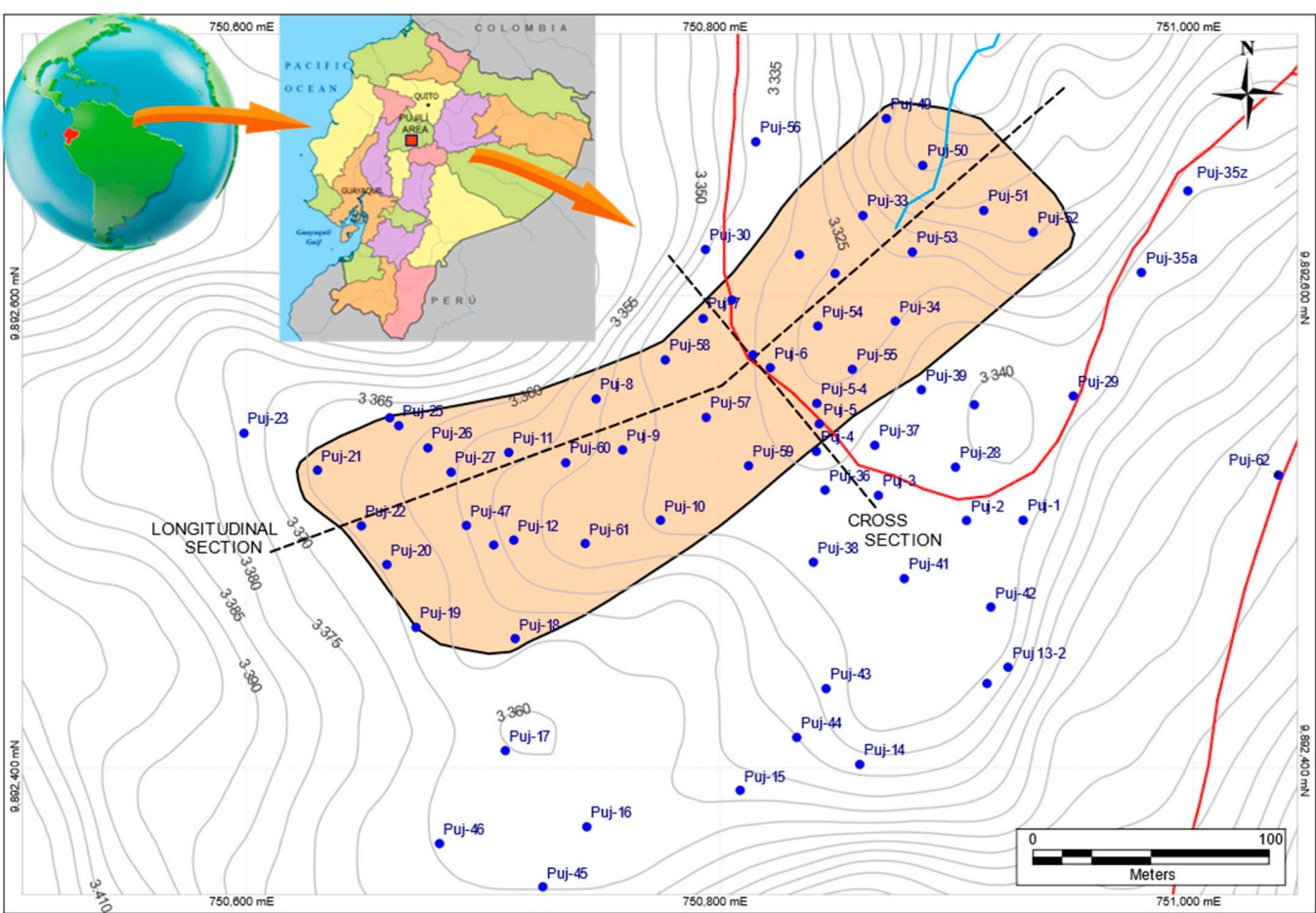

**Figure 1.** General situation map of the study area. The map shows a detailed plan view of the position of every HVSR-based single-station survey (blue dots) applied to the studied moving-landslide area (shaded) and its surroundings. Roads are depicted as red lines, and water courses as blue lines. The two interpretation sections that are analyzed (marked as the longitudinal section and cross-section) are displayed in black dashed lines.

This investigation uses the HVSR technique to determine and delineate the location of the rupture zone in this medium-sized landslide located in Pujilí canton (Ecuador).

## 2. Geographical Setting and Geological Framework

The study area is located in the Pujilí canton (Cotopaxi province, Ecuador), on the eastern slope of the Western Andean Cordillera, close to the Cotopaxi volcano (Figure 1). In the Cachi Alto community (751200 E/8982230 N UTM coordinates, 17S zone, WGS84 datum in the central point of the area), the local geomorphology is characterized by a steep relief with slopes ranging from moderate (8°–16°) to steep (26°–45°).

The Patoa River crosses the study area as the main watercourse, and various other small-size courses (referred to by Ecuadorians as *quebradas*) concur in this zone. Generally, the river has a dendritic morphology. The area's average elevation is 3330 m above sea level (m.a.s.l.) [36]. The meteorology is semi-humid, with annual rainfall between 1000 and 2000 mm, and a High-Mountain Equatorial climate type [33,36]. An outstanding

geomorphological feature of the area is the Punteras hill, located to the SE of the studied area, which corresponds to a dacitic dome formation that dominates the site (3515 m.a.s.l.).

The Central University of Ecuador conducted geological studies in this area [33,34]. The Río Cala Unit was identified as a Pre-Quaternary unit located from the NE zone to the SE of the Cachi Bajo community. Additionally, the NE of Cachi Alto was characterized as the rocky basement of the entire area, composed mainly of basaltic andesites (Figure 2).

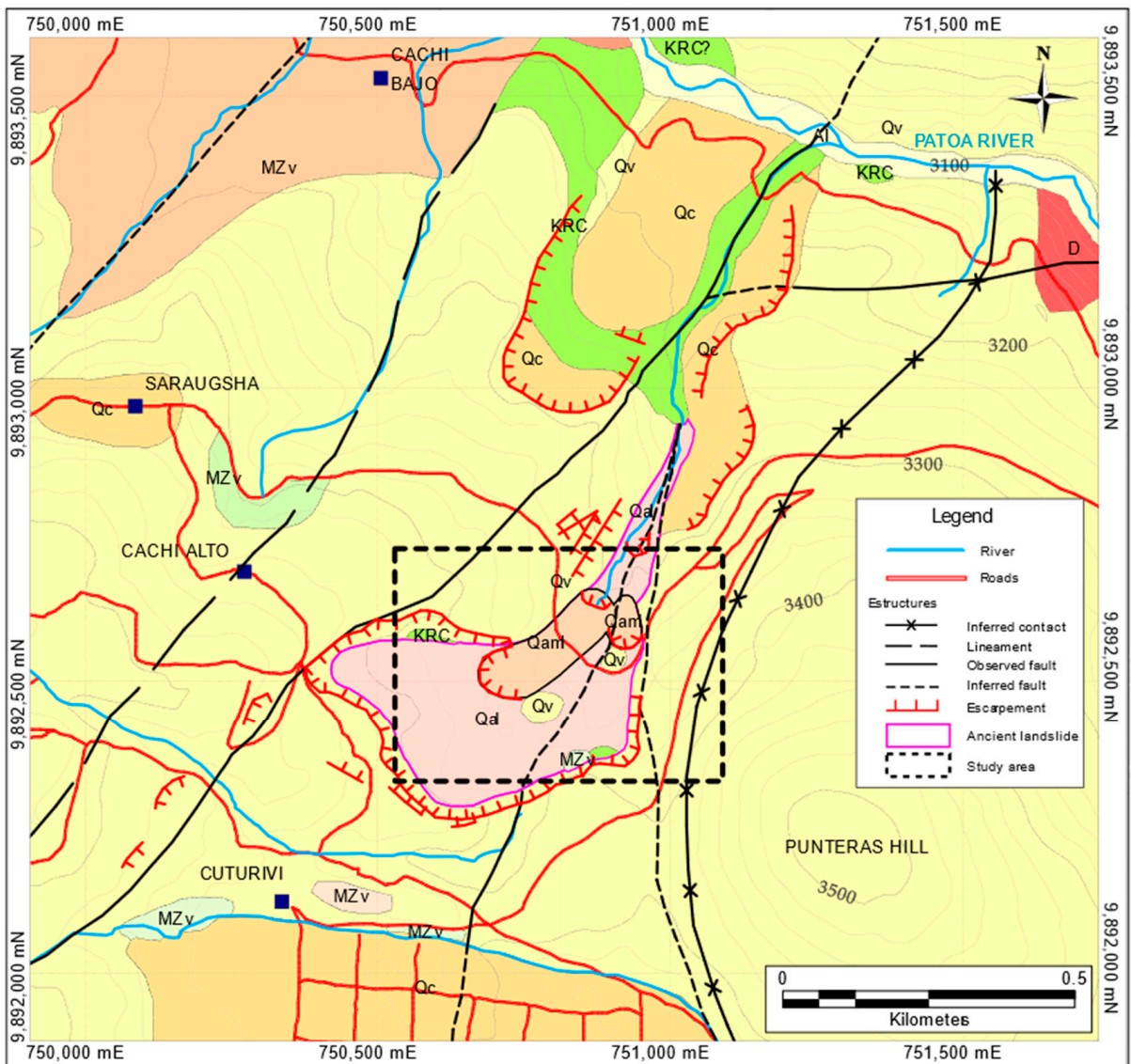

**Figure 2.** Geological map of the landslide area under study (dashed black square). Abbreviations: Mesozoic Zumbagua Group (MZv), Cretacic Rio Cala Unit (KRC), Cangahua Formation (Qv), colluvial (Qc), colluvial ancient landslide (Qal), colluvial actual medium-sized landslide (Qaml), alluvial (Al), dacitic igneous intrusion (D). The yellow-shaded area is the main landslide studied, shown in Figure 1. Modified from [34].

The Zumbagua Group has a varied stratigraphy, and it outcrops in the Cachi Bajo community area, with a wide extension. While geological gaps were observed in the NE of Cachi Alto, outcrops of sandstones and siltstone outcrops appeared in other nearby areas. Similarly, shales occur to the NW of Cuturiví Grande, and conglomeratic breccias appear to the NE. Small outcrops of pyroxenic andesites are also observed at the ends of the main landslide area, located SE of Cachi Alto and NE of the Cerro Punteras hill [33,34].

The recent Quaternary deposits cover the majority of the area under study. They are composed of materials from the Canguahua Formation, which are compact and slightly cemented sediments of volcanic origin (tuffs and ashes). The presence of recent and old colluvial deposits over these materials is related to previous and ancient landslides in the area. Finally, the presence of alluvial sediments is related to the Patoa River's embedded basin, running towards the north side of the site [33] (Figure 2).

The landslide under study was identified as being of medium magnitude and found within a larger ancient landslide (8.3 Ha in size), triggered by the 1966 $M_w$ 5.7 Pujilí earthquake, affecting the upper area (Figure 2). Supposedly, the activation of the studied landslide began in 2018 and started to affect directly and visibly the local paved access road to the Cachi Bajo sector in 2019. The landslide-investigation zone covers an area that is more than 300 m long and with an average width of 65 m [35,36]. Previous investigations [33] proposed that this landslide has a co-seismic nature and is associated with active faults in the NNE–SSW and NE—SW directions, which are part of the Pallatanga–Pujilí–Calacalí regional system, and/or with the Tambillo fault. However, studies on the *cangahua* materials and stability analyses conducted on the largest and oldest landslide concluded that the new landslide could be related to water saturation in the area [35]. All the material geomechanical parameters involved are similar, except for the density, which is lower in the shallow and altered *cangahua*.

Slope deformations and deep-seated gravitational-slope deformations were identified [34], confirming the activity of the landslide under study. In contrast, the previous large-magnitude landslide is now inactive (due to the aging process). Additionally, investigations underline the possible formation of a broad pull-apart basin as a general feature of the area (Figure 3).

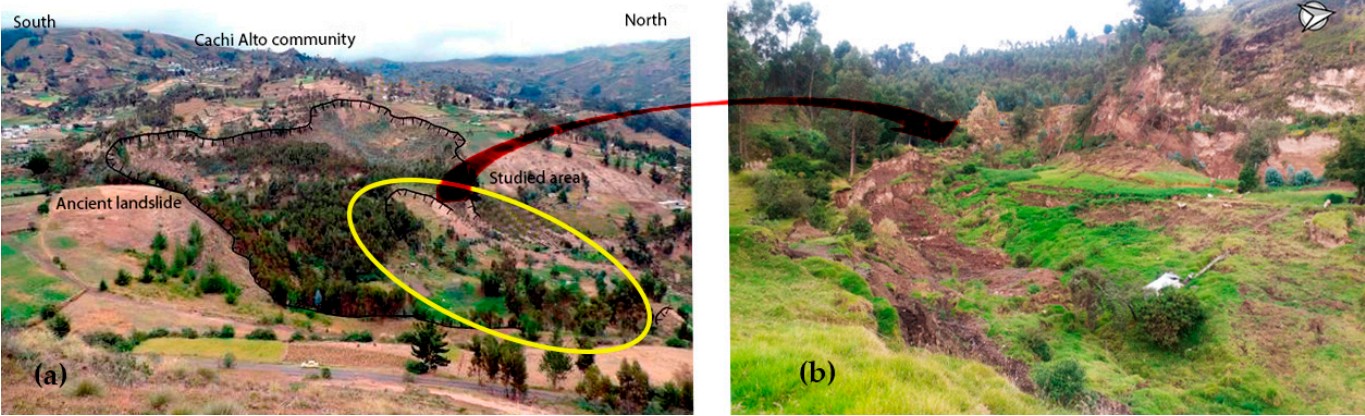

**Figure 3.** General views of the area under study. (**a**) The ancient landslide, with its head marked by a black line with ticks; the studied area is marked with a yellow oval (a car can be seen as a scale at the bottom of the picture), modified from [34]. (**b**) Studied landslide area viewed from the middle to the top (note the rest of the destroyed house, white roof in the right center of the landslide).

Some geophysical investigations were also conducted in the nearby town of Pujilí [37]. The results demonstrated general fundamental frequency peaks ($f_o$) associated with values between 1.0 and 1.5 Hz and a sedimentary overburden between 1.5 and 3.0 Hz. However, higher-frequency peaks appeared in two sets: from 5.0 to 7.0 Hz and from 7.0 to 10.0 Hz, with the H/V ratio equal to or greater than these fundamental frequencies (in the same peak category). These second peaks (sometimes prominent) were related to a surface layer of possibly alluvial materials (loose and not compact).

## 3. Background and Methods

### 3.1. Geophysical Methods

Geophysical research applied to landslide studies is widely recognized, with seismic and electrical methods being the most valuable techniques for such investigations [7–9]. The movement of the mass and its sliding rupture generate changes in geophysical parameters that can be used to analyze and recognize a landslide body or even study its dynamic motion [31]. In this research, HVSR passive surveys are applied as the main investigation tools, along with two active seismic techniques to define and obtain a basic starting model and provide complementary data.

Nakamura [16] defined the relationship between the thickness of sediments overlaying a basement through the natural vibration frequency in a two-layer model. This was based on the hypothesis that the vertical component of ambient noise in the sediment's surface ground has the characteristics of the source and is relatively influenced by Rayleigh waves on the sediment. Thus, the vertical component may be removed both from the source and from the Rayleigh-wave effects from the horizontal components to clearly determine the fundamental vibration mode in the sedimentary layer [16,38].

The fundamental frequency $f_o$ of a soft sediment layer overlying a harder layer (considered as the basement) can be more accurately evaluated using thickness-averaged sediment-shear-wave velocity $V_s$. Nakamura [15,16] demonstrated experimental results from ambient-vibration and velocity profiles (analyzed through down-hole drilling) that confirmed that the amplitude of the HVSR peak is related to $V_s$ contrast between the upper layer of soft sediments and the more rigid underlying layer. Thus, the maximum or peak amplitude increases proportionally to the $V_s$ ratio of the two layers. In some cases, a higher secondary peak may appear related to other changes in elastic impedance (i.e., velocity times a layer material density), as indicated by [39,40]. The geological materials must exhibit a high-level elastic parameter contrast to establish $f_o$ in a two-layer model (sediment—rock or a soft-soil layer over a compact or cemented half-space), meaning a marked difference in seismic impedance must characterize them. Currently, the HVSR technique is recognized for setting $f_o$ and fundamental periods of the ground [41–45], defining sediment thickness [18,19,46], and studying soil–structure interactions [15,47].

When using the HVSR technique, the thickness of the soft materials ($H$) from the acquisition of the values of the natural frequency of soil ($f_o$) can be defined by applying Equation (1) [15,16]:

$$H = \frac{nV_s}{4f_o} \tag{1}$$

where $n$ is the vibration mode, without attenuation or irregularity, and $V_s$ is the sediment's shear-wave velocity averaged over the whole sedimentary soft material above the basement. The concept of the fundamental resonant frequency of a sedimentary layer over a basement and its implied amplification factor in the HVSR single-station measure is widely accepted by most authors [41,45].

The use of the dispersion or ellipticity curve to design inversion models for defining the $V_s$ distribution in the ground is not considered an accurate tool. This is because infinite solutions can be obtained that satisfy the ellipticity-curve adjustment [18,48]. However, stratigraphic knowledge or geophysical and geotechnical parameters related to the investigation area can be used to build an initial model that allows the ellipticity curve's inversion (using these constrained conditions). In this way, a response model assessing $V_s$ in the ground can be defined and fitted [18,49]. Constrained models are applied in the interpretation and marker identification at the frequency $f_o$, which is related to the material column to be determined. This limits the number of models that can be obtained, and the results are fitted by the ground stratigraphy associated with the thickness of materials and $V_s$ [49,50].

The maximum frequency is widely accepted as a reliable diagnostic tool for subsurface conditions at a given site. It was observed that independent site conditions can considerably influence the amplitude of the spectral ratio (H/V). The noise wavefield's composition

and direction include different proportions of surface waves (Love and Rayleigh) and body waves [51]. Amplification caused by dynamic conditions (such as earthquakes) on topographic slopes is more significant in specific directions than in others. Furthermore, it relates to the azimuthal variation observed in microtremor measurements [52]. The azimuth-parameter value has been considered to be related to the directionality inside the ground (directivity) [29,53]. The directional effects in landslides have been demonstrated, and there is evidence that this vibration directionality is perpendicular to the tectonic structures that control the area [54,55].

Directional analyses performed on HVSR data provide knowledge about how landslide bodies received contributions from ambient-noise energy along the three spatial axes. As the amplification in the moving ground on the two horizontal components can be different, there is more significant movement in specific azimuths, i.e., a directional amplification is produced [56]. The presence of fractures or cracks in sliding-material bodies can also show directionality in HVSR logs (especially in prominent fault zones). These discontinuities can generate the presence of directional amplification, particularly those related to large-scale open cracks [55–57] or even microcracks [26].

Seismic refraction and MASW surveys have previously been used to define the $V_s$ of shallow materials and build initial ground models. Seismic refraction is a geophysical technique that makes it possible to obtain geometrical distribution profiles of compression or primary wave velocities ($V_p$) along extensions of receivers (geophones). A field of elastic waves is created by applying energization (an explosion or a falling mass) to the ground, which is critically refracted in the different layers of the subsoil [58,59]. The MASW technique is an active seismic survey that allows the study of the distribution of $V_s$ through geophone alignment (profile) coupled on the ground surface [60,61]. The MASW technique is based on the net-dominance factor of Rayleigh waves, which are related to $V_s$. Thus, the $V_s$ velocity distribution is analyzed from the phase-velocity dispersion records, obtaining a one-dimensional model (1D). The data processing involves the inversion of wave-dispersion data to obtain a $V_s$ distribution up to 40 m [62,63].

A $V_{s\,sed}$ average can be calculated from the $V_s$ values of soil and soft sedimentary layers through the surface to the basement or compact material depth. This procedure is similar to the $V_{s30}$ calculation used in seismic-ground-profile-classification codes [64,65]. Nevertheless, instead of only accounting for the first 30 m, the thickness of all the sediments over the compact substrate is considered, following Equation (2):

$$V_{s\,sed} = \frac{\sum_{i=1}^{N} h_i}{\sum_{i=1}^{N} \frac{h_i}{V_{si}}} \qquad (2)$$

where $V_{s\,sed}$ is the shear-wave-velocity average to be considered, $N$ is the total number of layers, and $h_i$ and $V_{si}$ are each layer's thickness and the shear-wave velocity, respectively.

### 3.2. Research Methodology

In the present work, no direct complementary investigations or other data such as drilling surveys (boreholes) or laboratory material tests, were available for the study area. Therefore, the definition of the landslide-rupture surface was based on the application and analyses of the HVSR technique, complemented by use of active seismic profiles as geophysical seismic techniques (seismic refraction and MASW). Interpreting the active seismic surveys applied to profiles based on $V_p$ and $V_s$ distribution in each geological layer was the first step in defining an initial ground model. The data provided by the HVSR single-station surveys were used to obtain the ellipticity curve (dispersion), $f_o$, and spectral ratio H/V (also known as amplification, $A_o$) at each point. These values enable calculation of the position of the surface rupture (thickness of soft material), $K_g$ index, and the azimuth of the natural vibration by analyzing its directivity.

The rupture surface was calculated using three different approaches: (i) the $V_s$ value obtained from active seismic techniques; (ii) the inversion of the ellipticity curve to obtain

a $V_s$ vertical distribution of geological materials; and (iii) a $V_s$ average value from the inversion results. The first methodology (i) employed the traditional Nakamura [16] formulation to relate the shear-wave velocity ($V_s$) of sedimentary materials and its associated $f_o$ value. This approach used the multichannel analysis of seismic waves (MASW) profile technique to investigate the $V_s$ value of the sedimentary material. The second methodology proposed (ii) used the results of the ellipticity-curve inversion at each HVSR station point measured inside the landslide. The inversion was performed on an initial constrained model defined from the active seismic-profiling survey data (refraction and MASW) and provided a vertical distribution of $V_s$ velocities. The last methodology (iii) utilized the average $V_s$ value obtained from the previous inversion results as input to the Nakamura formulation (i.e., the third methodology combined the first and second methodologies).

The three approaches were used to define the rupture surface, and their differences were evaluated to assess their application capabilities. Additionally, the directionality of the natural vibration was analyzed (microtremor) [26,29,56,66,67] and applied to establish internal rupture zones or their relationship with the sliding mass. The distribution of the vulnerability index ($K_g$) parameter, which indicates the potential capacity of earth masses to remain unstable [15], was also analyzed. The methodology presented in this work can be applied to other areas with similar geological characteristics, where constraining conditions can be defined to adjust or use the different analyzed methods.

### 3.3. Seismic-Active-Techniques Surveys

A seismic refraction profile, 69 m in length with 24 reception channels and geophones spaced 3 m apart (applying five energization points), was carried out, reaching an exploration depth of approximately 30 m. The survey was conducted in the central area of the landslide, where the movement began. This enabled observation of the best relationship between $V_p$ values of the mobilized and the fixed material, thus defining $V_p$ impedance. Moreover, this area is more stable and has a flatter surface than the sliding area. The profile extended over the access road and crossed an incipient fracture zone that can be observed on the surface between PUJ-4 and PUJ-2 (see Figure 1). The results included $V_p$ values for each layer, defining the characteristics and distribution of the different geological materials in the area (depth and geometry of the interfaces along the profile as a 2D model).

The active MASW investigation was performed with 24-channel equipment and applied to a single profile over the exact alignment where the seismic refraction survey was executed. Using the same array as the refraction survey, the data obtained provided the $V_s$ values and distribution of layers in depth. Five 2.4-s-long stacked records were measured and transformed by fast Fourier transform (FFT) to the frequency domain. The obtained curves on the fundamental mode were inverted to reach a final $V_s$ distribution model. As the MASW technique does not make it possible to obtain a 2D distribution (as in the case of seismic refraction), the analysis of all records (channels) from a unique alignment was applied to the center's profile with a 1D distribution [60]. Results displayed $V_s$ values for the defined layers within the first 35 m.

### 3.4. Seismic Passive-HVSR-Technique Surveys

An irregular mesh of points was established and distributed inside and outside the sliding-mass area. The former belonged to the mobilized materials and involved 40 single-station surveys (3 were not inside, but close to it). At the same time, the latter was considered fixed ground and involved 30 single-station points. For every 70 HVSR single-station surveys performed, at least 20 min of vibration noise was recorded. The three-geophone equipment was oriented towards magnetic north when placed on the ground. Since a directivity analysis was later conducted, the equipment (horizontal-component geophones) at every station was oriented in the N–S and E–W directions.

For processing HVSR data, the free software, GEOPSY [68], developed by the SESAME project, was used [69,70]. From the measured data, composed of the three vibration components (N–S, E–W, and vertical) records, a discretization of the total time of the

wavelets was made in 25-s windows. The FFT was applied after the remotion of the transients, and after incorrect signals were filtered and eliminated. Once FFT was used, the horizontal components (N–S, E–W) were geometrically composed to obtain a single horizontal value and perform the spectral-ratio analysis (H/V). Smoothing of 40% was applied using a filter to obtain the ellipticity curves [71].

The records obtained in HVSR single-station tests may provide more reliable data for determining the $V_s$ at each measured point [21,40]. Nevertheless, considering a constrained interpretation model of their results (ellipticity curve from HVSR records), $V_s$ distribution can be consistently defined under the tested points [21,50]. To conduct the inversion procedure, starting ellipticity curves were analyzed using free DINVER software, part of GEOPSY package [68–70]. An initial ground model is required, with velocity and thickness limited as the main layer parameters. The input model must consider a range of values for each layer's depth and velocity. The software employs a conditional neighborhood algorithm to perform inversion and optimization through successive iterations [72,73]. The final model obtained (with the minimum possible adjustment error) shows the $V_s$ distribution values for each proposed layer proposed and its thickness. These layers must be categorized as either loose or soft compacted sediments, or high compact or cemented sediments, with a boundary established at $V_s$ over 600 m/s [16].

The DINVER module [73] facilitated the analysis of the directional effects in HVSR data processing by rotating the components of the spectral ratio H/V obtained at each measurement site. The representation was generated in steps of 10°, covering half the circumference amplitude (from 0° N to 180° N), with spectral amplitude values plotted as contours on the distribution of the function of frequency and direction of movement [74,75]. The north direction was represented by 0°, while the south direction was represented by 180° (orientation direction of the N–S horizontal-component equipment). The results from 180° to 360° were symmetrical, expressing the direction of the directivity.

### 3.5. Vulnerability Index ($K_g$)

The vulnerability index $K_g$ [16] was defined for a specific efficiency percentage related to $A_o$ and $f_o$ obtained in a HVSR-measurement survey. If a 60% efficiency and a shear-wave velocity for the basement of 600 m/s are considered, Equation (3) can be used to obtain $K_g$:

$$K_g = \frac{A_o^2}{f_o} \tag{3}$$

This index indicates the ease of material deformation and defines the ground's weakness. The index is also related to the potential capacity of the instability of a landslide: according to Nakamura's investigations, $K_g$ values exceeding ten are associated with shear-strain deformation [15,16].

It should be noted that $A_o$ values may vary over an extended period during the collection of data sets. Typically, the stability of this parameter is usually valid within one-month periods [76]. In this research, the data set was collected during two periods of one-month duration, fulfilling the premise of the invariability of this index. Other parameters typically do not vary significantly over short periods.

## 4. Results and Analysis of the Data

### 4.1. Seismic Refraction: Compressional Velocity ($V_p$) Model

The processing of the seismic-refraction-profile data provided a distribution of four geophysical levels to the first 30 m (see Figure 4). The materials' distribution began with a discontinuous surface layer of with an average thickness of approximately 2 to 4 m, exhibiting a $V_p$ of 369 m/s. This layer was related to soft soil or loose surface materials. The next geophysical level, showing a variable geometry, was related to highly altered soils or significantly altered *cangahua* with low-to-medium compaction ($V_p$ = 590 m/s), and a thickness ranging from 1 to more than 10 m.

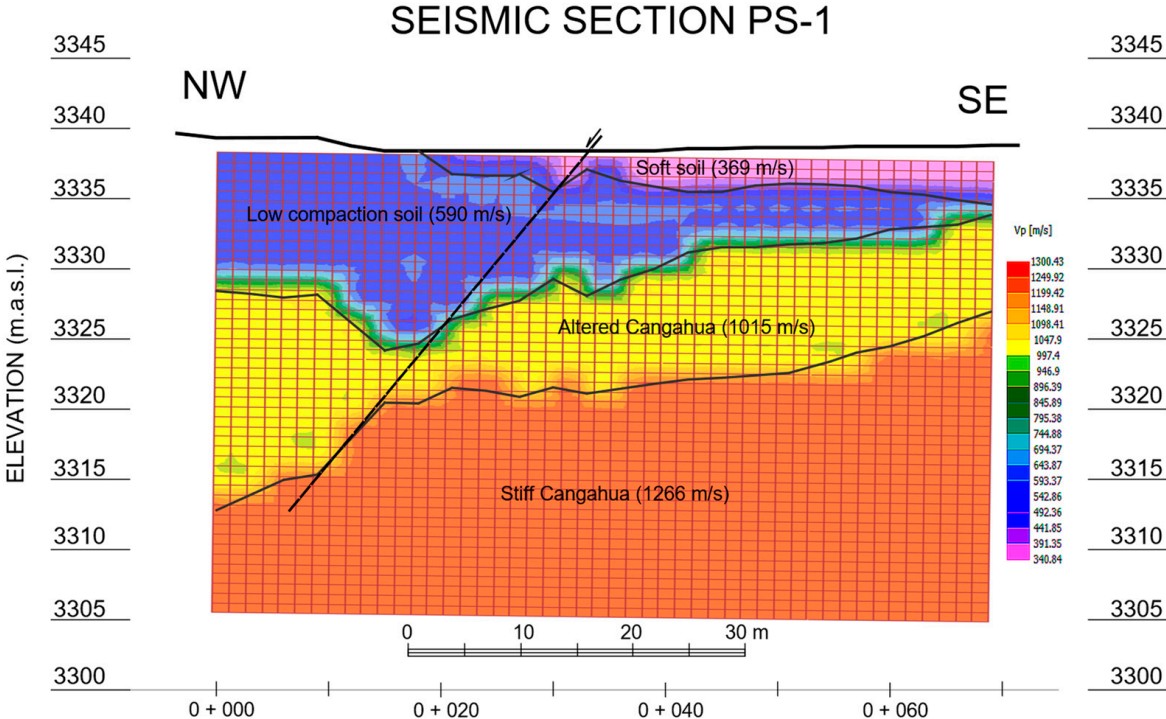

**Figure 4.** Seismic-refraction-interpretation section. The $V_p$ averaged values and interpreted geophysical levels are defined. From the center of the profile to the NW end, it is possible to identify a prone fault (black dashed line and arrow pointing to the movement) related to the landslide.

The last two layers corresponded to the *cangahua* material. This was the predominant material on the surface of the investigated area and was defined by two velocity levels: a shallow level, with $V_p$ = 1015 m/s, extending from the bottom of the previous level to approximately 20 m in depth, and the infinite semi-space, where $V_p$ = 1266 m/s. Higher $V_p$ values indicated high compaction and the possible cementation of this material compared to the previous *cangahua* level, which had a higher alteration grade and, consequently, presented less hardness. A significant impedance contrast of $V_p$ values was observed between the shallow materials (the first and second geophysical levels), which did not have high compaction, and the substratum materials (the third geophysical level). The impedance-contrast ratio was about 2.74, which allowed the differentiation of these levels in the HVSR surveys.

In the layered geometrical interpretation, a jump can be observed in the deeper materials towards the beginning of the profile (NW zone in Figure 4). This may have been related to a fault structure detected on the surface.

### 4.2. MASW Technique: Shear Velocity ($V_s$) Distribution and Constrained-Model Definition

The interpretation results of the MASW profile technique provide a vertical distribution model of $V_s$ and its corresponding material thickness set to the center of the whole geophone alignment, resulting in a one-dimensional distribution (1D) [60].

The results demonstrated a superficial layer with a $V_s$ of 142 m/s and a thickness of 2.50 m. This layer was followed by a geophysical level 6.53 m thick (up to a depth of 9.0 m) and with a $V_s$ of 241 m/s. These two superficial levels overlay a deeper level with a $V_s$ of 465 m/s, which had a thickness of 10.40 m. This means that the deepest point of this bottom interface was 19.4 m. The previous sequence was located over a layer with a $V_s$ of 612 m/s (Figure 5). As $V_s$ > 600 m/s, this last layer corresponded to the compact or cemented *cangahua* material, and it can be considered immobile or fixed in the landslide zone [16].

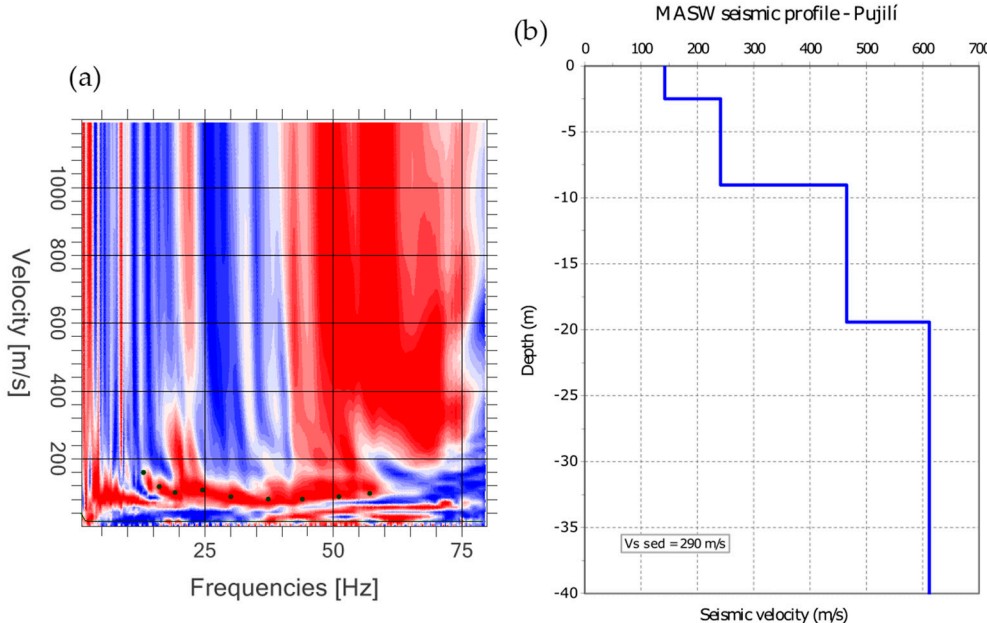

**Figure 5.** Dispersion-graph data (**a**) and distribution of $V_s$ (**b**) as a result of the MASW seismic-survey interpretation. This interpretation was conducted on the same area as the application of the seismic-refraction profile (see Figure 1). The $V_{s\,sed}$ value for shallow sedimentary materials (average from $V_s < 600$ m/s) is also shown.

Based on these results, Equation (2) was used to calculate the average shear-wave velocity for the surface materials (the first shallow soft levels overlaying the compact materials' substratum or basement materials), which was found to be $V_{s\,sed} = 290$ m/s. This value was subsequently used to interpret the HVSR-test results and determine the $V_s$ of the sliding sediments by applying Equation (1).

The impedance contrast in the S-waves between the shallow sediments and the compacted materials for the proposed model was 2.89, a value similar to that obtained in the seismic-refraction P-wave velocity. This value allowed the definition of an initial two-layer model.

Therefore, the constrained model to be considered in the further inversion procedure of the ellipticity curve was proposed to consist of the five geophysical levels (GL), shown in Table 1, where $V_s$ and thickness are defined within a specific interval to limit the constructed models. The $V_s$ interval was determined with the central value close to that specified in the MASW-technique-based model.

**Table 1.** The values of the proposed constrained geophysical model used for the ellipticity curve inversion.

| Geophysical Levels | Velocity Interval ($V_s$) | | Thickness | |
|---|---|---|---|---|
| | From (m/s) | To (m/s) | From (m) | To (m) |
| GL-1 | 80 | 350 | 2.0 | 10.0 |
| GL-2 | 150 | 500 | 3.0 | 20.0 |
| GL-3 | 250 | 700 | 5.0 | 30.0 |
| GL-4 | 250 | 900 | 10.0 | 100.0 |
| GL-5 | 350 | 1500 | - | - |

The GL-1 level was related to shallow materials, such as soft soils and highly weathered *cangahua*. At the GL-2 level, the geological materials necessarily had more compact characteristics than the materials in the previous layer, and the weathered *cangahua* had a lower degree of alteration. The material defined in the GL-3 level could have been the last layer involved in the sliding motion, reaching the separation limit from the fixed material,

since the $V_s$ ranged from 250 to 700 m/s (for static and compact *cangahua*, $V_s > 600$ m/s was assumed). The GL-4 material may have been related to the compacted and less altered *cangahua*, as part of the fixed material. The last level (GL-5) was beyond the range and was used to control the inversion process (it was not considered in further calculations); it had no thickness because it was a semi-infinite space.

*4.3. HVSR Data Results: Natural Frequency ($f_o$) Determination*

Figure 6 shows the processing results of the selected points on which single or two predominant peaks were set.

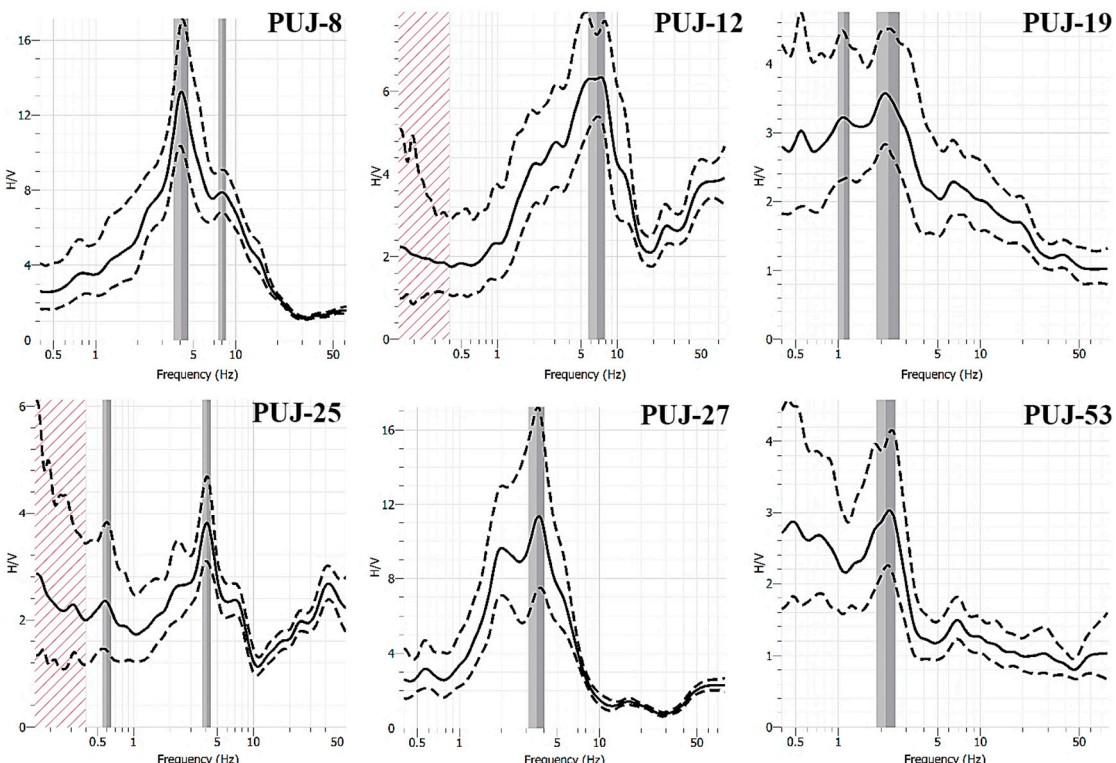

**Figure 6.** Six selected examples of HVSR-data-processing results (ellipticity curves). The continuous black line is the mean value of the H/V ratio (the amplification $A_o$), and the black dashed lines are the standard deviation of this value. The center of the gray bands indicates the position of the applied fundamental frequencies $f_o$ and the extent of the standard deviation.

Most of the ellipticity curves obtained in this study (80%) presented a single clear peak, with the highest amplitude corresponding to the primary or fundamental frequency. The remaining curves exhibited either a secondary peak (with a lower amplitude, and not considered in the calculations) or a broad peak.

The primary peak was analyzed and defined as the $f_o$ value, corresponding to the frequency with the highest spectral ratio (H/V) or amplification ($A_o$). Some of the curves exhibited a single and clear narrow peak (such as PUJ-8, PUJ-12, or PUJ-27) with associated minor peaks, while others had broad peaks, such as PUJ-9 (Figure 6). The $f_o$ value was extracted as the maximum value, with more significant spectral ratio or amplification for that frequency. This is indicated by the middle of the two gray bars. The continuous black line represents the ellipticity curve, while the dashed line shows the standard deviation (Figure 6).

Peaks with significant amplitudes and clear definitions are typically associated with the presence of a large impedance contrast [51,69]. Conversely, curves that lack clear peaks (broad), or have a secondary peak, are related to internal changes in the layers or interfaces.

A flat curve corresponds to shallow basement (hard rock) areas or highly compact and cemented-material layers [69].

Table 2 shows the fundamental ground-vibration frequency ($f_o$) values for all the single-station HVSR surveys. The amplification ($A_o$) value is also shown for each fundamental frequency, along with the $K_g$ calculated value from Equation (3). The $f_o$ frequency ranged from 9.59 Hz to 1.22 Hz, with an average of 3.16 Hz and a standard deviation of 1.58 Hz. The amplification $A_o$ ranged from 13.24 to 1.67 (dimensionless), with an average of 3.83 and a standard deviation of 2.06. Note that these intervals and averages corresponded to the total 70 HVSR measured points, but they exhibited minimal variations if only the surveyed points on the sliding mass were considered. The $K_g$ value ranged from 42.97 at its maximum to a minimum of 0.79, with an average of 6.07 and a standard deviation of 7.01 (also dimensionless).

**Table 2.** Values of the data obtained in the HVSR-processing records. Indication of the fundamental frequency ($f_o$), the amplification of the spectral ratio ($A_o$). The calculated value of the vulnerability index ($K_g$) from Equation (3) is given. The last two values are dimensionless.

| Point | $f_o$ (Hz) | $A_o$ | $K_g$ | Point | $f_o$ (Hz) | $A_o$ | $K_g$ |
|---|---|---|---|---|---|---|---|
| PUJ-1 | 3.31 | 2.80 | 2.37 | PUJ-35z | 2.33 | 3.06 | 4.02 |
| PUJ-2 | 4.08 | 3.35 | 2.75 | PUJ-36 | 12.50 | 3.49 | 0.97 |
| PUJ-3 | 3.94 | 2.41 | 1.47 | PUJ-37 | 1.84 | 2.71 | 3.99 |
| PUJ-4 | 5.51 | 4.67 | 3.96 | PUJ-38 | 3.45 | 2.43 | 1.71 |
| PUJ-5 | 4.75 | 1.94 | 0.79 | PUJ-39 | 2.00 | 5.06 | 12.80 |
| PUJ-6 | 2.43 | 5.44 | 12.18 | PUJ-40 | 3.60 | 7.31 | 14.84 |
| PUJ-7 | 2.15 | 2.33 | 2.53 | PUJ-41 | 1.75 | 1.67 | 1.59 |
| PUJ-8 | 4.08 | 13.24 | 42.97 | PUJ-42 | 1.73 | 2.81 | 4.56 |
| PUJ-9 | 2.60 | 3.90 | 5.85 | PUJ-43 | 3.34 | 4.03 | 4.86 |
| PUJ-10 | 2.58 | 3.47 | 4.67 | PUJ-44 | 6.03 | 4.64 | 3.57 |
| PUJ-11 | 2.91 | 2.62 | 2.36 | PUJ-45 | 1.63 | 2.79 | 4.78 |
| PUJ-12 | 2.25 | 3.54 | 5.57 | PUJ-46 | 1.78 | 2.80 | 4.40 |
| PUJ-13 | 2.34 | 2.18 | 2.03 | PUJ-47 | 3.00 | 3.03 | 3.06 |
| PUJ-14 | 4.26 | 4.89 | 5.61 | PUJ-48 | 2.22 | 3.08 | 4.27 |
| PUJ-15 | 7.06 | 3.78 | 2.02 | PUJ-49 | 3.12 | 4.28 | 5.87 |
| PUJ-16 | 1.51 | 3.28 | 7.12 | PUJ-50 | 9.59 | 14.82 | 22.90 |
| PUJ-17 | 1.73 | 2.43 | 3.41 | PUJ-51 | 6.97 | 5.58 | 4.47 |
| PUJ-18 | 1.59 | 4.16 | 10.88 | PUJ-52 | 5.85 | 2.35 | 0.94 |
| PUJ-19 | 6.76 | 6.30 | 5.87 | PUJ-53 | 2.56 | 2.68 | 2.81 |
| PUJ-20 | 2.90 | 6.70 | 15.48 | PUJ-54 | 2.09 | 3.60 | 6.20 |
| PUJ-21 | 3.84 | 3.79 | 3.74 | PUJ-55 | 1.78 | 3.76 | 7.94 |
| PUJ-22 | 4.84 | 10.27 | 21.79 | PUJ-56 | 2.84 | 3.49 | 4.29 |
| PUJ-23 | 3.98 | 3.12 | 2.45 | PUJ-57 | 2.17 | 2.98 | 4.09 |
| PUJ-24 | 2.15 | 2.85 | 3.78 | PUJ-58 | 2.31 | 2.30 | 2.29 |
| PUJ-25 | 4.08 | 3.80 | 3.54 | PUJ-59 | 1.80 | 2.85 | 4.51 |
| PUJ-26 | 2.33 | 5.20 | 11.61 | PUJ-60 | 2.31 | 2.54 | 2.79 |
| PUJ-27 | 3.57 | 11.17 | 34.95 | PUJ-61 | 1.93 | 3.35 | 5.81 |
| PUJ-28 | 4.01 | 2.43 | 1.47 | PUJ-62 | 2.10 | 2.81 | 3.76 |
| PUJ-29 | 2.00 | 2.68 | 3.59 | PUJ5-4 | 3.71 | 2.50 | 1.68 |
| PUJ-30 | 2.41 | 2.48 | 2.55 | PUJ6-4 | 2.20 | 4.06 | 7.49 |
| PUJ-31 | 3.24 | 1.91 | 1.13 | PUJ7-2 | 2.13 | 1.88 | 1.66 |
| PUJ-32 | 2.17 | 5.50 | 13.94 | PUJ13-2 | 2.30 | 2.23 | 2.16 |
| PUJ-33 | 2.18 | 4.81 | 10.61 | PUJ-200 | 2.34 | 3.93 | 6.60 |
| PUJ-34 | 5.40 | 2.28 | 0.96 | PUJ-201 | 2.01 | 2.24 | 2.50 |
| PUJ-35a | 2.41 | 3.01 | 3.76 | PUJ-202 | 1.22 | 4.28 | 15.02 |

*4.4. Ellipticity-Curve Inversion*

The constrained model presented in Table 1 served as a starting point to initiate the modeling of the iterations of the inversion. However, these iterations were not necessarily linked (as fixed limits) to the model (which was free and used as a limitation for the

adjustment iterations). The process described in [77,78] was followed to eliminate errors and ensure accuracy in the final results.

Once this model was used as the initial input in the software, the final model for each HVSR -survey ellipticity curve was determined through a series of iterations. The inversion -iteration process of the ellipticity curve converged to a group of models with different adjustment values (referred to as misfit in the software).

An example of the different model results is shown in Figure 7 for the PUJ-57 HVSR point, where broadly different adjustment misfit curves are displayed (with a color scale from 1.5% to 10.0%, Figure 7a). The iteration results needed to converge into a reliable model until the minimum misfit (adjust) value was reached (represented in Figure 7b as a $V_s$ vertical distribution vs. depth).

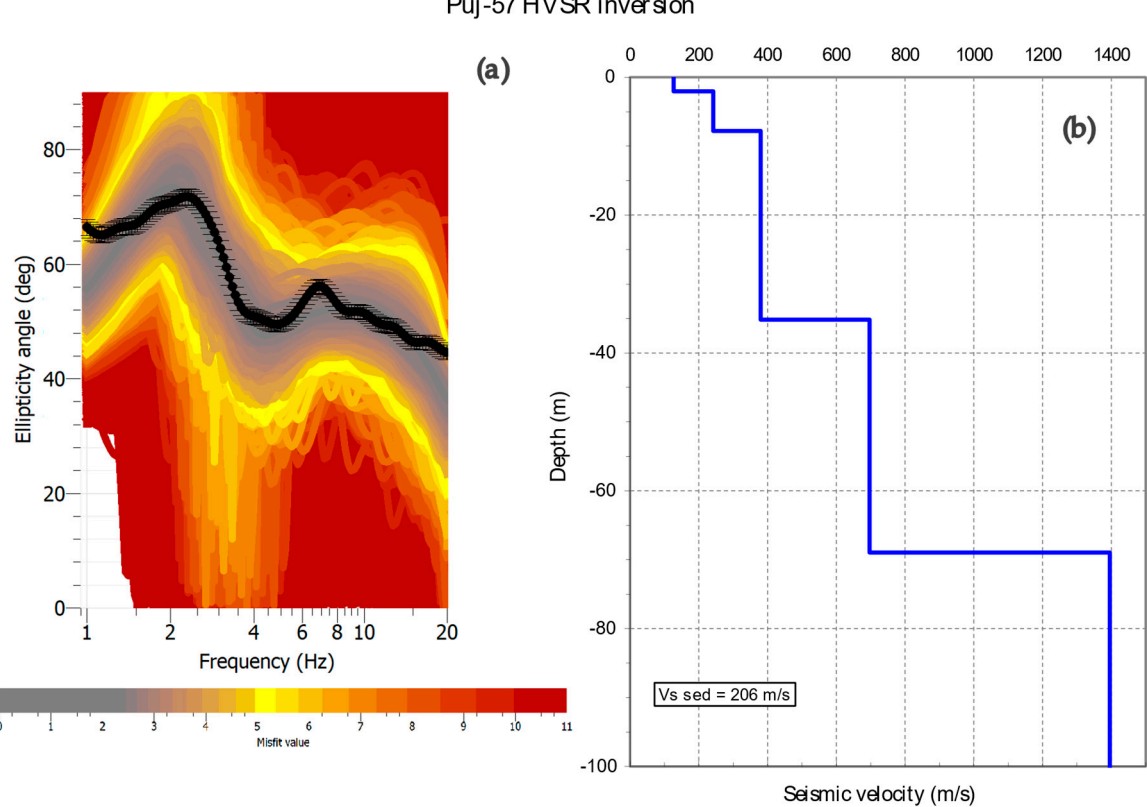

**Figure 7.** Example of a HVSR ellipticity-curve inversion for PUJ-57 point. (**a**) Ellipticity curve (using a black dotted line) inversion graphical results of all adjusted curves and their misfit value (shown using a color scale). (**b**) Vertical $V_s$ distribution of layers versus depth as the obtained solution after computing the inversion procedure. The $V_s$ average value of the sediments over the compact *cangahua* (considered as the basement) obtained is shown ($V_{s\,sed}$).

The inversion was carried out only on the HVSR points measured over the sliding mass, i.e., where the rupture surface was analyzed. The complete inversion results for the $V_s$ at each of the 37 station points inside the sliding-mass area are shown in Table 3. The values of $V_s$ for the five geophysical levels defined and the $V_{s\,sed}$ obtained from Equation (2) are listed. The average values for each layer and $V_{s\,sed}$ are also provided. It can be observed that the L4 layer representing the compacted *cangahua* had a value of over 600 m/s [15]. In comparison, the $V_s$ in the L5 layer was an almost constant value (~1400 m/s) because that layer was used to control the stability of the models. The $V_{s\,sed}$ average of all the considered points was 317 m/s, a value close to that obtained previously for the MASW-model interpretation (290 m/s).

**Table 3.** Values of shear-wave velocity obtained from the inversion of the ellipticity curve of the proposed model (L# $V_s$ is the shear-wave velocity obtained at each geophysical level).

| Hvsr Point | L1 $V_s$ (m/s) | L2 $V_s$ (m/s) | L3 $V_s$ (m/s) | L4 $V_s$ (m/s) | L5 $V_s$ (m/s) | $V_{s\,sed}$ (m/s) |
|---|---|---|---|---|---|---|
| PUJ-4 | 119 | 289 | 400 | 718 | 1354 | 289 |
| PUJ-5 | 134 | 339 | 458 | 710 | 1466 | 322 |
| PUJ-6 | 103 | 249 | 450 | 474 | 1467 | 305 |
| PUJ-7 | 110 | 292 | 502 | 740 | 1481 | 307 |
| PUJ-8 | 113 | 289 | 372 | 683 | 1496 | 257 |
| PUJ-9 | 107 | 254 | 473 | 770 | 1496 | 331 |
| PUJ-10 | 112 | 295 | 395 | 732 | 1496 | 277 |
| PUJ-11 | 109 | 273 | 415 | 697 | 1409 | 360 |
| PUJ-12 | 115 | 281 | 477 | 747 | 1467 | 334 |
| PUJ-20 | 109 | 284 | 577 | 793 | 1096 | 389 |
| PUJ-21 | 114 | 270 | 387 | 747 | 1424 | 296 |
| PUJ-22 | 110 | 289 | 369 | 725 | 1496 | 266 |
| PUJ-23 | 115 | 307 | 612 | 726 | 1409 | 391 |
| PUJ-25 | 124 | 316 | 612 | 754 | 1496 | 396 |
| PUJ-26 | 117 | 301 | 487 | 793 | 1481 | 319 |
| PUJ-27 | 117 | 239 | 383 | 676 | 1481 | 267 |
| PUJ-30 | 117 | 281 | 477 | 817 | 1467 | 357 |
| PUJ-31 | 133 | 304 | 419 | 676 | 1438 | 307 |
| PUJ-32 | 109 | 281 | 424 | 711 | 1496 | 303 |
| PUJ-33 | 107 | 286 | 554 | 711 | 1467 | 375 |
| PUJ-34 | 113 | 275 | 533 | 770 | 1496 | 354 |
| PUJ-36 | 105 | 264 | 308 | 576 | 1166 | 231 |
| PUJ-47 | 124 | 295 | 473 | 643 | 1438 | 304 |
| PUJ-49 | 117 | 289 | 317 | 697 | 1452 | 249 |
| PUJ-50 | 100 | 273 | 522 | 683 | 1496 | 342 |
| PUJ-51 | 120 | 307 | 560 | 785 | 1481 | 344 |
| PUJ-52 | 115 | 284 | 492 | 725 | 1467 | 324 |
| PUJ-53 | 127 | 275 | 354 | 565 | 1409 | 278 |
| PUJ-54 | 113 | 292 | 527 | 817 | 1481 | 385 |
| PUJ-55 | 114 | 275 | 380 | 612 | 1481 | 256 |
| PUJ-57 | 133 | 267 | 383 | 588 | 1251 | 206 |
| PUJ-58 | 149 | 333 | 450 | 754 | 1424 | 368 |
| PUJ-59 | 124 | 313 | 380 | 690 | 1467 | 284 |
| PUJ-60 | 135 | 304 | 468 | 793 | 1452 | 367 |
| PUJ-5-4 | 112 | 286 | 324 | 631 | 1496 | 256 |
| PUJ-6-4 | 114 | 292 | 543 | 718 | 1481 | 357 |
| PUJ-7-2 | 122 | 320 | 492 | 697 | 1481 | 330 |
| **AVERAGES** | **117** | **288** | **453** | **707** | **1441** | **317** |

*4.5. Thickness Calculation*

The MASW seismic technique was used to determine the separation between the soil overburden and the altered *cangahua* over the compact substratum, showing $V_{s\,sed}$ = 290 m/s for this sedimentary material. Furthermore, from the ellipticity-curve inversion, a difference value that exceeded it by 27 m/s ($V_{s\,sed}$ = 317 m/s) was obtained. The impedance-contrast value obtained from the elastic waves (up to 2.7 on average) led to the separation of the altered and soft materials over the cemented or compact basement of the rupture surface based on this change.

Table 4 shows the thicknesses of surface sediments obtained from both $V_s$ values and using Equation (1), which ranged from 5.40 to 0.60 m. The average difference between the two procedures was computed as 2.46 ± 0.96 m.

**Table 4.** Values of the frequency ($f_o$) and the thickness of materials (m) according to the MASW analysis (TH-1 column) and for the $V_{s\ sed}$ average value from the ellipticity-curve inversion (TH-2 column) in relation to the separation with the immobilized substrate. The points within the investigated active landslide are marked in red.

| Hvsr Point | $f_o$ (Hz) | TH-1 (m) | TH-2 (m) | Hvsr Point | $f_o$ (Hz) | TH-1 (m) | TH-2 (m) |
|---|---|---|---|---|---|---|---|
| PUJ-1 | 3.31 | 21.9 | 23.9 | PUJ-35z | 2.33 | 31.1 | 34.0 |
| PUJ-2 | 4.08 | 17.8 | 19.4 | PUJ-36 | 12.50 | 5.8 | 6.3 |
| PUJ-3 | 3.94 | 18.4 | 20.1 | PUJ-37 | 1.84 | 39.4 | 43.1 |
| PUJ-4 | 5.51 | 13.2 | 14.4 | PUJ-38 | 3.45 | 21.0 | 23.0 |
| PUJ-5 | 4.75 | 15.3 | 16.7 | PUJ-39 | 2.00 | 36.3 | 39.6 |
| PUJ-6 | 2.43 | 29.8 | 32.6 | PUJ-40 | 3.60 | 20.1 | 22.0 |
| PUJ-7 | 2.15 | 33.7 | 36.9 | PUJ-41 | 1.75 | 41.4 | 45.3 |
| PUJ-8 | 4.08 | 17.8 | 19.4 | PUJ-42 | 1.73 | 41.9 | 45.8 |
| PUJ-9 | 2.60 | 27.9 | 30.5 | PUJ-43 | 3.34 | 21.7 | 23.7 |
| PUJ-10 | 2.58 | 28.1 | 30.7 | PUJ-44 | 6.03 | 12.0 | 13.1 |
| PUJ-11 | 2.91 | 24.9 | 27.2 | PUJ-45 | 1.63 | 44.5 | 48.6 |
| PUJ-12 | 2.25 | 32.2 | 35.2 | PUJ-46 | 1.78 | 40.7 | 44.5 |
| PUJ-13 | 2.34 | 31.0 | 33.9 | PUJ-47 | 3.00 | 24.2 | 26.4 |
| PUJ-14 | 4.26 | 17.0 | 18.6 | PUJ-48 | 2.22 | 32.7 | 35.7 |
| PUJ-15 | 7.06 | 10.3 | 11.2 | PUJ-49 | 3.12 | 23.2 | 25.4 |
| PUJ-16 | 1.51 | 48.0 | 52.5 | PUJ-50 | 9.59 | 7.6 | 8.3 |
| PUJ-17 | 1.73 | 41.9 | 45.8 | PUJ-51 | 6.97 | 10.4 | 11.4 |
| PUJ-18 | 1.59 | 45.6 | 49.8 | PUJ-52 | 5.85 | 12.4 | 13.5 |
| PUJ-19 | 6.76 | 10.7 | 11.7 | PUJ-53 | 2.56 | 28.3 | 31.0 |
| PUJ-20 | 2.90 | 25.0 | 27.3 | PUJ-54 | 2.09 | 34.7 | 37.9 |
| PUJ-21 | 3.84 | 18.9 | 20.6 | PUJ-55 | 1.78 | 40.7 | 44.5 |
| PUJ-22 | 4.84 | 15.0 | 16.4 | PUJ-56 | 2.84 | 25.5 | 27.9 |
| PUJ-23 | 3.98 | 18.2 | 19.9 | PUJ-57 | 2.17 | 33.4 | 36.5 |
| PUJ-24 | 2.15 | 33.7 | 36.9 | PUJ-58 | 2.31 | 31.4 | 34.3 |
| PUJ-25 | 4.08 | 17.8 | 19.4 | PUJ-59 | 1.80 | 40.3 | 44.0 |
| PUJ-26 | 2.33 | 31.1 | 34.0 | PUJ-60 | 2.31 | 31.4 | 34.3 |
| PUJ-27 | 3.57 | 20.3 | 22.2 | PUJ-61 | 1.93 | 37.6 | 41.1 |
| PUJ-28 | 4.01 | 18.1 | 19.8 | PUJ-62 | 2.10 | 34.5 | 37.7 |
| PUJ-29 | 2.00 | 36.3 | 39.6 | PUJ-5-4 | 3.71 | 19.5 | 21.4 |
| PUJ-30 | 2.41 | 30.1 | 32.9 | PUJ-6-4 | 2.20 | 33.0 | 36.0 |
| PUJ-31 | 3.24 | 22.4 | 24.5 | PUJ-7-2 | 2.13 | 34.0 | 37.2 |
| PUJ-32 | 2.17 | 27.3 | 36.5 | PUJ-13-2 | 2.30 | 31.5 | 34.5 |
| PUJ-33 | 2.18 | 33.3 | 36.4 | PUJ-200 | 2.34 | 31.0 | 33.9 |
| PUJ-34 | 5.40 | 13.4 | 14.7 | PUJ-201 | 2.01 | 36.1 | 39.4 |
| PUJ-35a | 2.41 | 30.1 | 32.9 | PUJ-202 | 1.22 | 59.4 | 65.0 |

### 4.6. Directivity (Azimuthal) Values

An example of six selected processing results is shown in Figure 8. These graphs represent the same information as an ellipticity curve, along with a distribution analysis of the vibration direction in relation to the azimuth (0° to 180°). The frequency values are on the X-axis, while the azimuthal-vibration direction is represented on the orthogonal Y-axis. The amplification or spectral ratio H/V is shown as a contour color scale.

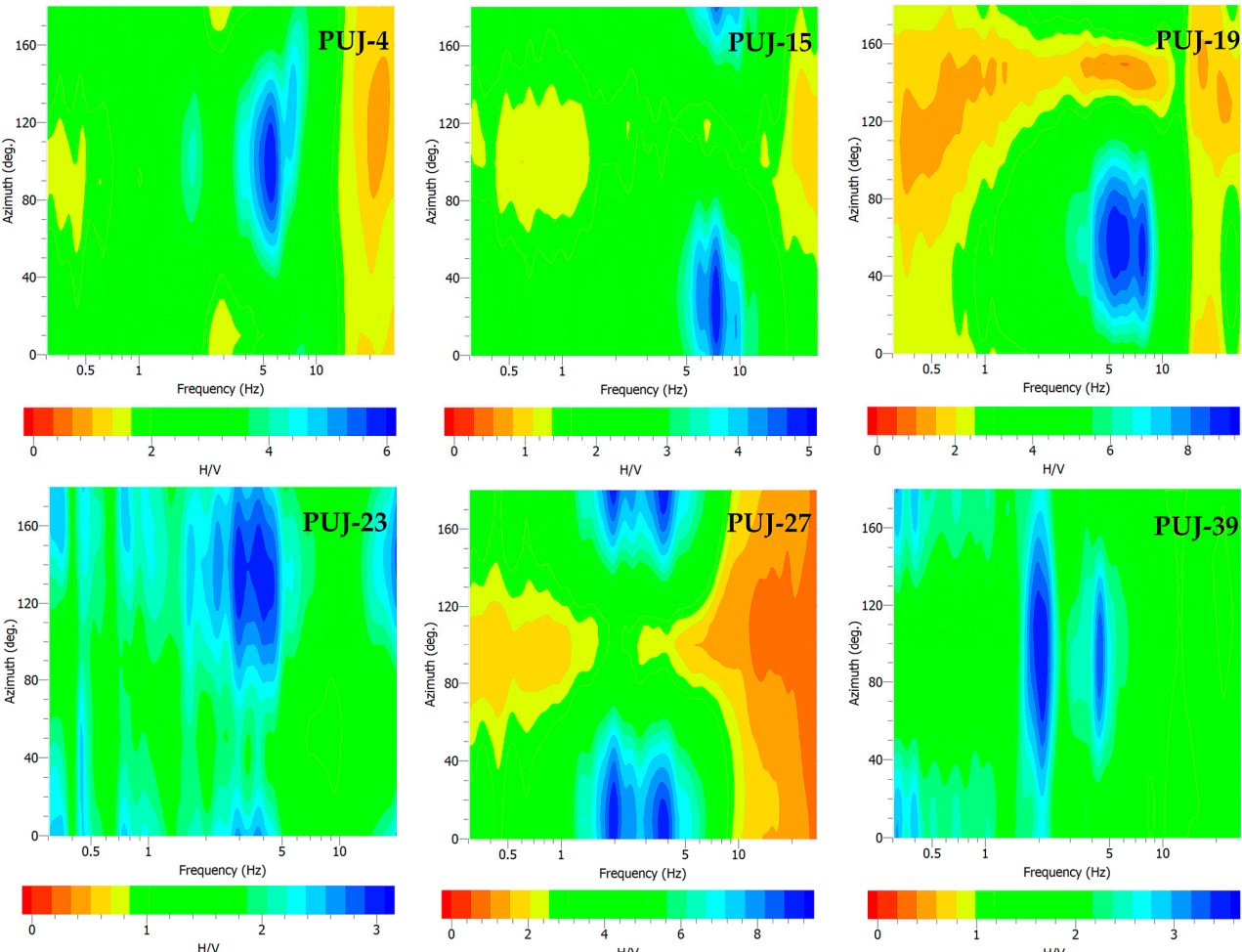

**Figure 8.** Directional analysis output graphic representation on six selected points, including clear single peaks (PUJ-4 and PUJ-15), broad peaks (PUJ-19 and PUJ-23), and peak curves (PUJ-27 and PUJ-39). Each figure shows the contour plots of the HVSR curve as a function of frequency (X-axis) and rotation angle (Y-axis) in the direction of magnetic north. The color scale represents the HVSR amplitude.

The results showed one or two peaks of the HVSR-ellipticity curve, marked by the high amplification value (blue-band-color contoured intervals are used on these scales, with the contouring scale varying on every graph). Two examples of each case are shown: broad peaks, compared to clear dominant peaks, and a double-peak appearance. The azimuthal directivity value assigned to each HVSR single-point station was computed from the maximum value in these graphs. It can be observed that the maximum values in the directivity graphs were isolated, indicating that the directivity analysis was related to the upper level and not directly to the substratum [29].

Table 5 presents the values obtained for each HVSR survey point measured in the area. The direction was calculated as the value perpendicular to the azimuth, which was expected to correspond to the general trend of the internal structures [10,13,21]. The directivity graphs in Figure 8 only consider half of the circumference because the values were symmetric [69]. Therefore, the main direction was computed and shown in Table 5 to represent these values as a whole. In addition, the general declination value for the area was taken into consideration.

**Table 5.** Values of the directionality of the HVSR measurements. Indication of the main direction (azimuth column) in degrees, considering north as 0° and the value of the direction of the structures (direction column) calculated orthogonally to the azimuth.

| Hvsr Point | Azimuth (°) | Direction (°) | Hvsr Point | Azimuth (°) | Direction (°) |
|---|---|---|---|---|---|
| PUJ-1 | 313 | 223 | PUJ-35z | 333 | 243 |
| PUJ-2 | 268 | 178 | PUJ-36 | 338 | 248 |
| PUJ-3 | 269 | 179 | PUJ-37 | 292 | 202 |
| PUJ-4 | 278 | 188 | PUJ-38 | 341 | 251 |
| PUJ-5 | 317 | 227 | PUJ-39 | 277 | 187 |
| PUJ-6 | 317 | 227 | PUJ-40 | 272 | 182 |
| PUJ-7 | 51 | 141 | PUJ-41 | 14 | 104 |
| PUJ-8 | 6 | 96 | PUJ-42 | 264 | 174 |
| PUJ-9 | 54 | 144 | PUJ-43 | 343 | 253 |
| PUJ-10 | 10 | 100 | PUJ-44 | 26 | 116 |
| PUJ-11 | 11 | 101 | PUJ-45 | 76 | 166 |
| PUJ-12 | 58 | 148 | PUJ-46 | 53 | 143 |
| PUJ-13 | 6 | 96 | PUJ-47 | 2 | 92 |
| PUJ-14 | 58 | 148 | PUJ-48 | 99 | 189 |
| PUJ-15 | 25 | 115 | PUJ-49 | 119 | 209 |
| PUJ-16 | 30 | 120 | PUJ-50 | 79 | 169 |
| PUJ-17 | 17 | 107 | PUJ-51 | 20 | 110 |
| PUJ-18 | 79 | 169 | PUJ-52 | 289 | 199 |
| PUJ-19 | 56 | 146 | PUJ-53 | 12 | 102 |
| PUJ-20 | 36 | 126 | PUJ-54 | 50 | 140 |
| PUJ-21 | 89 | 179 | PUJ-55 | 56 | 146 |
| PUJ-22 | 29 | 119 | PUJ-56 | 37 | 127 |
| PUJ-23 | 132 | 222 | PUJ-57 | 18 | 108 |
| PUJ-24 | 191 | 281 | PUJ-58 | 73 | 163 |
| PUJ-25 | 133 | 223 | PUJ-59 | 297 | 207 |
| PUJ-26 | 6 | 96 | PUJ-60 | 87 | 177 |
| PUJ-27 | 10 | 100 | PUJ-61 | 18 | 108 |
| PUJ-28 | 300 | 210 | PUJ-62 | 9 | 99 |
| PUJ-29 | 339 | 249 | PUJ-200 | 3 | 93 |
| PUJ-30 | 174 | 264 | PUJ-201 | 2 | 92 |
| PUJ-31 | 60 | 150 | PUJ-202 | 9 | 99 |
| PUJ-33 | 65 | 155 | PUJ 5-4 | 314 | 224 |
| PUJ-32 | 47 | 137 | PUJ-6-4 | 6 | 96 |
| PUJ-34 | 58 | 148 | PUJ-7-2 | 79 | 169 |
| PUJ-35a | 293 | 203 | PUJ-13-2 | 255 | 165 |

It should be noted that in the field, the sliding movements (internal and external) were associated with rupture processes (see Figure 9). Thus, Figure 10 displays the values from Table 5, which were represented on a map as an arrow pointing in the obtained azimuthal direction (geographical north was considered). For this representation, the values were adjusted to point towards the drift of the mobilization of the landslide. Figure 10 also includes a rose-diagram-distribution analysis of the frequency of the azimuth values for the 70 HVSR surveys applied over the studied area with the average direction (55°) of the sliding movement. The diagram depicts two main trends: 10° and 60° from magnetic north.

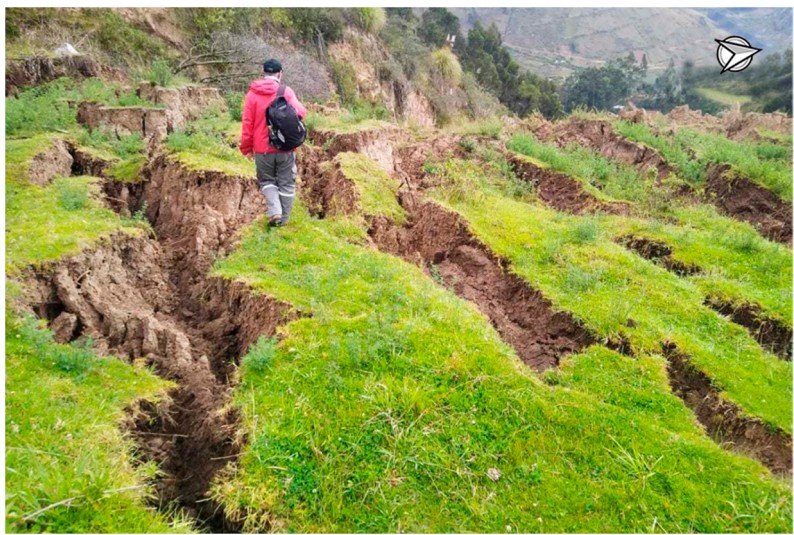

**Figure 9.** A picture view of the lower area of the landslide showing rupture structures (cracks) inside the sliding body mass. They are orthogonal to the direction of the sliding movement of the mass.

**Figure 10.** Plant-view representation of the azimuth of the measured points (blue arrows show parameters from Table 5, and the black arrow shows the moving direction). The investigated landslide is shaded dark, and slide-affected areas are light. Black dotted lines show the main fracture areas observed on the field. A rose diagram of azimuth frequency for all considered areas in the study is included (the black arrow shows the main sliding direction).

The directivity analysis of the HVSR-processed surveys in the active sliding zone was related to the internal structures of the materials, such as fractures and microcracks [29,56,67]. The analysis showed a relationship between these structures and the cracks observed on the surface. In this case, *cangahua* is a semi-compact material with stiff behavior. The directivity is related to the direction of the sliding of the surface material. The angle values obtained from the processing-output graphs (Figure 8) were related to shallow sediments and not basement materials because their influence area was around the $f_o$ value and did not extend across the 180 degrees represented on the graphs (which corresponds to the basement response [29]). Therefore, these directions (or their corresponding azimuths) mark the internal sliding-surface fracturing of the materials, which, depending on the type of movement in the area, corresponded to the compartmentalization of the sliding mass.

The HVSR points results measured outside the main landslide area show a direction towards this sliding zone (10°). This indicates that the landslide increased towards that area, as indicated by the light color in Figure 10. These structures were observed on the surface and were correlated with those shown in Figure 10 (black dotted lines). Outside the investigated landslide area, this value marks the prone structures in the sliding process, and the represented direction (Figure 10) shows the main sliding-mass process in a given course.

The magnitude of the movement can be observed in Figures 10 and 11, where the two parallel lines indicate the actual position of the road. During the landslide motion, there was a 60-m displacement of the road in the direction marked by the bold black arrow. The combined influence of topographic and geologic factors on the on-site-response directivity can explain the complexity of the spectral properties, with slightly diverging peaks at different frequencies related to various causal factors acting along similar directions [67].

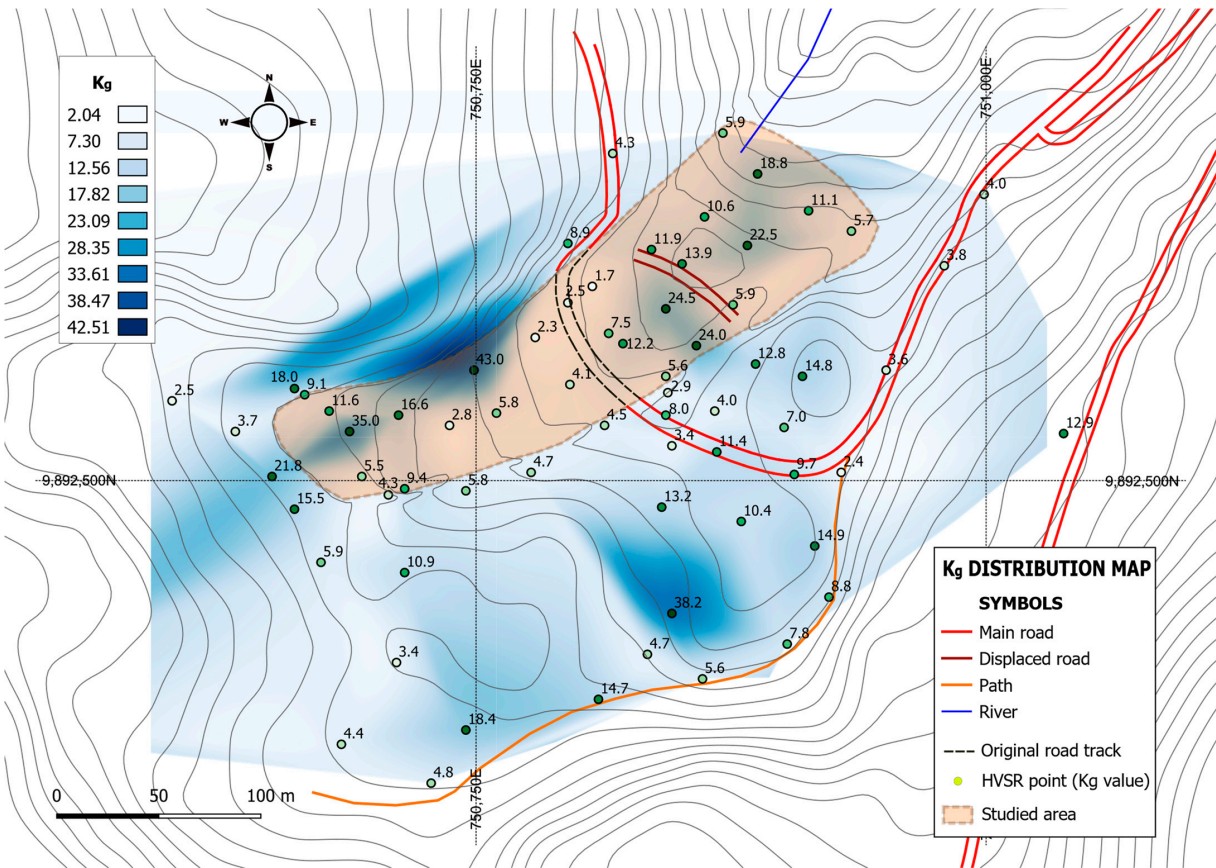

**Figure 11.** The $K_g$-index value-distribution map of the area under study was constructed by the TIN (triangular irregular network) interpolation method. Categorized representation with the index value for each surveyed point (see explanation in the text). The two parallel red lines inside the landslide area show the actual position of the road once it was displaced.

The most remarkable aspect of site amplification in landslide-prone slopes is its azimuthal variation. This means that the site response is amplified more significantly in specific azimuths than in other orientations [52]. Therefore, assessing the site seismic response (especially the azimuthal dependency) of unstable slopes and its relation with slope geometry and internal structure can provide more detailed knowledge about behavior during earthquakes, which is essential in landslide-hazard assessment [79].

*4.7. Vulnerability Index (Kg)*

Figure 11 represents the distribution of the $K_g$ parameter in the study area, constructed from the values in Table 2. It reveals the three sectors with the highest values indicating unstable conditions.

In these delimited zones, the $K_g$ value exceeded 30, corresponding to the actual maximum activity at the head and north sides of the landslide. The $K_g$ values between 10 and 25 were geometrically distributed from the head to the toe of the sliding mass, showing the most unstable areas of the phenomenon. The third zone, with $K_g$ over 30, was in a prone area outside the analyzed sliding mass. These areas were related to an incipient destabilization of materials, or to zones where the *cangahua* presented alterations with low levels of weathering. Additionally, high $K_g$ values were present along the central part of the landslide, which is now stable (see Table 2 and Figure 11).

Due to the assumed stiffness of the *cangahua* sliding materials, most of the obtained $K_g$ values were below the limit of ten [15], except for those in the indicated areas.

## 5. Discussion

This research proposed the investigation of the rupture surface in an active medium-sized landslide by applying the HVSR passive seismic technique and using seismic profiling (MASW and refraction) as a reference. The geological materials involved in the landslide were altered and stiff *cangahua*, which comprise the same geological material, both when static and when in motion. This makes the definition of the rupture surface complex, since their properties are similar, except for the degree of alteration (related to density). Nevertheless, a seismic-impedance contrast was defined near a value of 3.0 between the sliding and fixed materials, providing clear separation between the sliding materials to the static materials.

The $V_s$ velocity-distribution model was determined from the MASW-type passive seismic survey and adjusted by the information derives from the interpretation of the seismic-refraction survey. This gave an average $V_s$ value of 290 m/s for the moving materials involved in the sliding. It was used, along with the $f_o$ values obtained at every HVSR test point, to compute the thickness of the upper sedimentary layer over a compact layer based on Equation (1). The $f_o$ fundamental frequency used to define the depth of a basement is described in the works of various authors [10,79–81], including in Ecuador [19]. as a reliable tool for defining sediment thicknesses. It was also applied to obtain the landslide-rupture surface, as in [10,13,22,28,31].

In contrast to these studies, the moving and fixed masses in the actual investigation area were made of the same stratigraphic material. Therefore, the separation needed be based on the impedance changes observed due to weathering and alteration processes. Otherwise, it could have been unreliable [18,21].

The rupture-surface analysis based on the inversion of the ellipticity curve of the HVSR-measured data was defined by considering a constrained model [13,21]. The proposed starting model was based on the seismic-refraction and passive-MASW-survey results, in which the thickness and $V_s$ distribution were defined. An initial five-layer model was inverted until the best fit was reached in the adjustments of the field curve (ellipticity) and the theoretical curve (interpretation model) to the HVSR station points in the active sliding mass [68].

From the results of this inversion, a three-layer model was finally proposed, consisting of a shallow-material layer (containing the first two surficial soft-sediment layers), with a

low $V_s$ value, an intermediate layer related to a more compact material (altered *cangahua*-type materials), and a firm and hard *cangahua* basement (probably cemented). The average value of $V_s$ for the surface material ($V_{s\,sed}$) was calculated, yielding 317 m/s, corresponding to the sliding of the soft and altered materials over the compact *cangahua*. This value differed by 27 m/s from that obtained in the $V_{s\,sed}$ MASW survey (290 m/s). After introducing this new average $V_{s\,sed}$ value into Equation (1), a new rupture-surface depth was obtained at each HVSR station point, with an average difference of 2.46 m (ranging from less than 1.0 to 7.0) from the surface-rupture depth obtained using the $V_{s\,sed}$ MASW survey.

These results can be used to study the morphology of the landslide area and its rupture surface. Two interpretation sections over the landslide were defined (see Figure 1 for locations): a cross-section (Figure 12) and a longitudinal section (Figure 13). The three rupture surfaces obtained by applying the approaches were drawn in both representations. The values obtained for the mobilized sediment thickness (summarized in Table 4) were placed under each HVSR point, and the surface-rupture position was obtained by connecting them (with TH-1 represented by the dashed–dotted blue line and TH-2 by the red dashed line). Moreover, the material-stratigraphy analysis performed using the HVSR-inversion results from the three-layer model (shown as colored columns) also demonstrated a rupture surface, represented in these sections as a continuous black line.

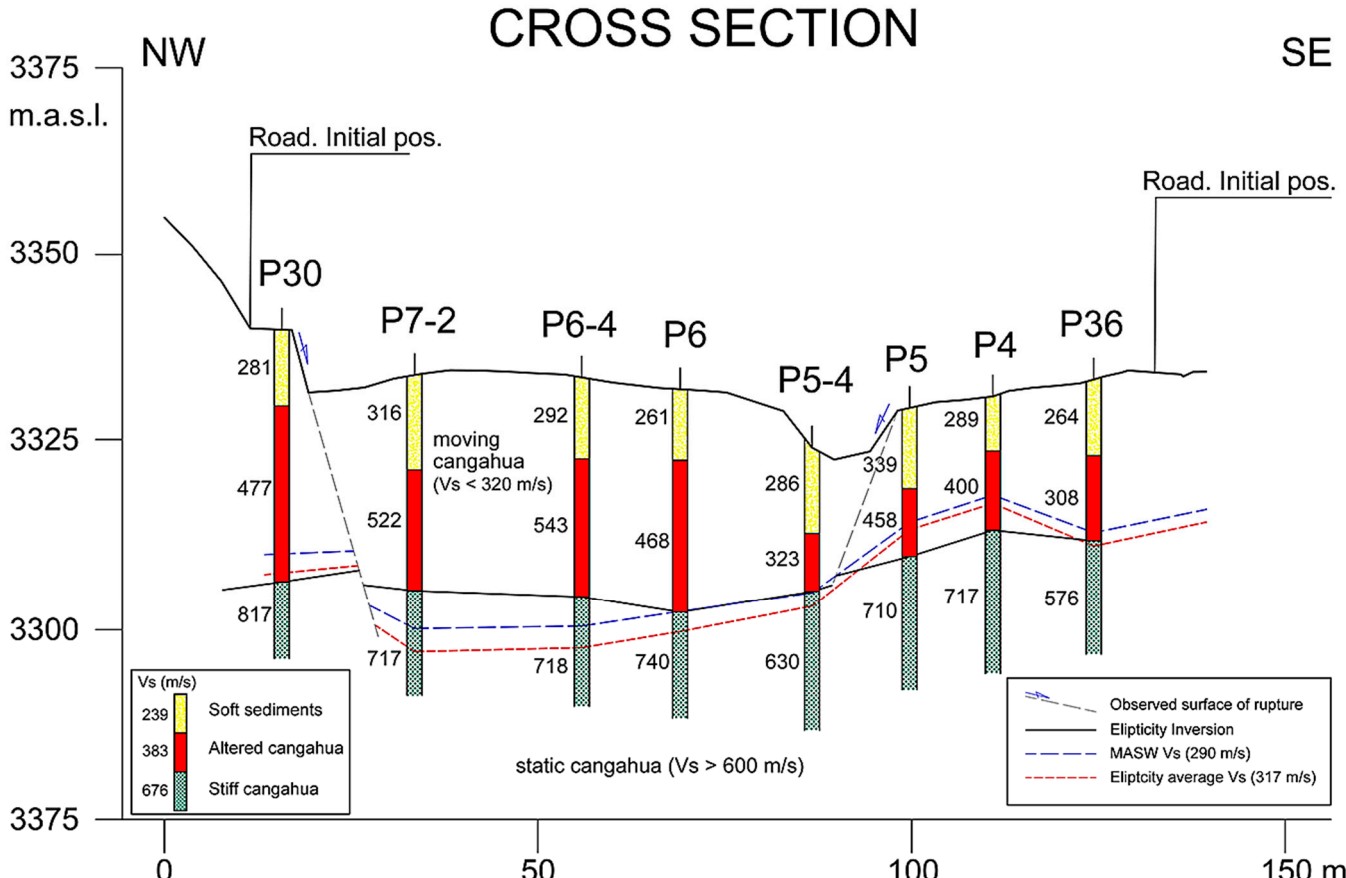

**Figure 12.** Cross-section of the landslide with the computed surface ruptures labeled. Colored columns represent the material distribution obtained from the inversion of the HVSR ellipticity curve. The three sliding surfaces are defined to compare: ellipticity-inversion curves, represented by continuous black line; computed MASW $V_s$ (<320 m/s), represented by blue dashed line; and $V_{s\,sed}$ average ellipticity value, represented by red dashed line. Two inferred faults are displayed in broad dashed black lines considering the delimitation of the landslide area. Arrows indicates movement.

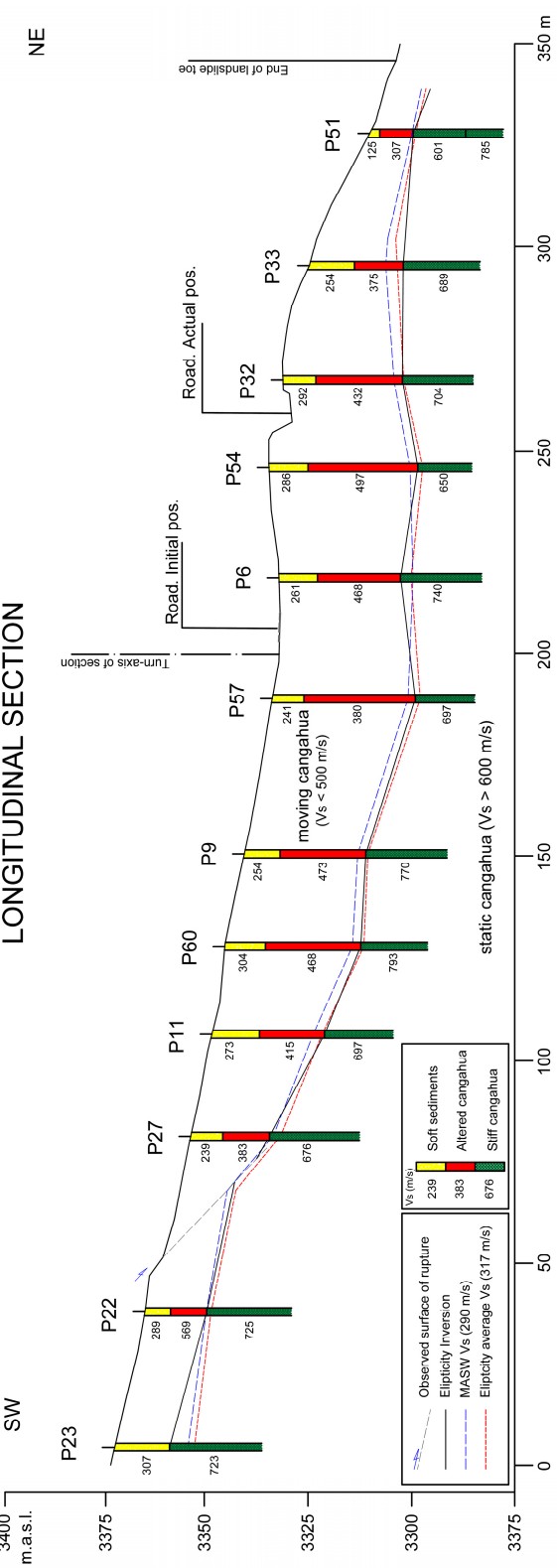

**Figure 13.** Longitudinal section of the landslide with the landslide-surface ruptures labeled. Colored columns represent the material distribution obtained from the inversion of the ellipticity curve. The three sliding surfaces are represented to compare: ellipticity-inversion curves, represented by continuous black line; computed MASW $V_s$, represented by blue dashed line; and ellipticity $V_{s\,sed}$, represented by red dashed line. An inferred fault is displayed by a broad dashed black line considering the delimitation of the top part of the landslide area (the arrow indicates the movement direction).

The cross-section drawn over the previous road layout shows a wide U-shaped geometry, in which the maximum sediment thickness appears near the NW flank (Figure 12). This is an active area of the landslide, where the flank fracture is more evident, with a 10-m subsidence (see the center and right images in Figure 3b). The longitudinal section (Figure 13) shows a shape that confirms the translational-landslide type that was initially postulated for this area, similar to that of the ancient landslide (see Figure 3a) [33].

The rupture surfaces obtained using the HVSR-ellipticity-inversion model and its associated average $V_s$ values are more closed and evident in the longitudinal section. However, they differ more in the cross-section (at points P-30, P-7-2, and P-6-4 HVSR). The maximum difference obtained between the surfaces set by the $V_s$ values related to the curve's inversion was 7.0 m. In contrast, the maximum difference was less than 3.0 m (2.46 m) between the levels, based on the two shear-wave velocities obtained.

It must be considered that the surface obtained from the ellipticity-curve inversion is an approximation that can be regarded as less precise than that obtained from the application of Equation (1). This is because the results depend on the accuracy of the inverted models and the impedance ratio between sliding and fixed materials. On the other hand, from the two $V_s$ velocity calculations based on Equation (1), even if the difference is low, the surface obtained from an average $V_s$ value on selected points inside the landslide can be considered more precise than that calculated from a general value (obtained from a unique MASW seismic profile).

A complete two-dimensional description of the sliding mass can be obtained by including all the HVSR points within the landslide area. However, a detailed digital elevation model (DEM) of the entire area should be created for a more accurate representation.

The obtained surface rupture and the results of the directivity and $K_g$-index analyses (Figures 10 and 11) provide a comprehensive understanding of the landslide process. The calculated sliding thickness can be used to estimate the volume of the involved moving mass, and the $K_g$ index can identify the potential areas of increased movement susceptibility (see Figure 11). The directivity analysis can also help to determine the internal structures of stiff materials (which can be cemented or highly compact, as in volcanic tuffs). Furthermore, it can be used to identify interior partitions or moving blocks, which can show different cinematic conditions.

The proposed HVSR-based survey methodology can be used as an early reference in the design of direct investigations, such as the drilling of boreholes and the determination of their depths, or the identification of better positions for conducting inclinometer-based monitoring of sliding masses.

It is worth noting that when collecting HVSR data, one common issue is the inclusion of anthropogenic environmental noise (e.g., noise related to industrial activities) because this tends to provide directional vibration noise. In the records, this anthropogenic noise is usually displayed by a sharp and clear peak in the ellipticity curve. The analyzed results can give "false" frequency values that mask the main fundamental frequency $f_0$ or confuse its definition. However, in this work, the study area had no anthropogenic activities or other vibrations during the data collection.

## 6. Conclusions

The application of an irregular mesh using the HVSR passive seismic technique as a stand-alone geophysical survey over a 19,000-m$^2$ active landslide and its surroundings provided ellipticity curves in which a main peak was exhibited. These peaks demonstrated the fundamental frequency of the ground $f_0$, and the related spectral ratio H/V (or amplification $A_o$).

As a reference, active seismic profiles were established using the refraction and MASW techniques, which allowed the design of P- and S-wave velocity distribution models and the determination of the geometrical shapes of the layers. The combination of both models allowed the establishment of an initial five-layer model, which was used to apply constrained inversion processing over the ellipticity curve.

The value assumed for the $V_{s\,sed}$ of the sliding materials was 290 m/s, which was obtained from the MASW survey, and it was close to that obtained from the inversion models of the ellipticity curves (317 m/s). The average seismic-impedance contrast (2.87) provided greater precision in the separation of the soft and altered sediments from compact materials. Therefore, the procedures can be considered to have similar levels of accuracy.

The three defined rupture surfaces, one of which was based on the inversion model from the HVSR-ellipticity curve and two of which were obtained by applying Equation (1) [15], differed in terms of their thickness values by less than 1.0 to 7.0 m (at their extremes) and had an average of 2.46 m. This indicates that all three rupture surfaces had a similar level of precision. The Nakamura relation in Equation (1) may be considered more useful due to the difficulties involved in inverting ellipticity curves, which can result in large errors.

The usefulness of the directivity analysis was demonstrated in this landslide area, where compact materials, such as the *cangahua* (stiff-to-rigid volcanic sediments) are present in the ground. The direction obtained (orthogonal to the HVSR azimuth angle) demonstrated the internal structural discontinuities in the sliding material. These fractures were related to the field-observed surficial fractures, and demonstrated blocks of sliding materials inside the landslide mass.

The $K_g$-based vulnerability-index analysis enabled the identification and delineation of areas with the potential for actual mobilization or that are prone to showing susceptibility to sliding in the environment. These were related to the top, center, and northwest landslide sectors, where the movement continued at slow rates (at points PUJ-27, PUJ-8, and PUJ-54), in addition to the surrounding area of PUJ-43, which exhibited a high $K_g$ value indicating a prone-to-slide area. The values in these areas were abnormal and exceeded the considered limit of ten. This is a potential indicator of the future persistence of a sliding motion or, as in the southeast area, zones that are prone to instability (some incipient cracks were observed in this zone).

Both of the methodologies analyzed provide an easy, quick, and low-cost approach to the study of landslide areas and demonstrate preliminary surface ruptures in their early phases, compared with investigations based on the drilling of boreholes. However, the impedance ratio between materials and the geological conditions of these materials should be considered to ensure accuracy. Overall, the HVSR technique demonstrated its usefulness as a preliminary exploration tool for landslide phenomena, enabling more detailed geological and geotechnical investigations (e.g., designing the depth to be used in a hole-drilling campaign, or the use of monitorization with inclinometers).

The application of this HVSR-based passive seismic geophysical technique and methodology can be used to complement the three-dimensional investigation of landslides by using a regular grid of single station points along the mass and surroundings of a landslide. Such investigations can provide fundamental information regarding the depth of the competent (immobile) substrate, the position of the rupture surface, and the determination of the volume of the sliding mass in movement, in addition to traditional geological and geotechnical investigations (e.g., boreholes and geophysics).

A potential future investigation would involve correlating these stand-alone geophysical investigations with perforations and monitoring data to adjust and specify both the obtained results and the changes due to materials and environmental conditions.

**Author Contributions:** Conceptualization, O.A.-P., D.B. and F.J.T.; methodology, O.A.-P.; software, O.A.-P. and D.B.; validation, O.A.-P., D.B., F.J.T. and J.G.-R.; formal analysis, O.A.-P. and F.J.T.; investigation, O.A.-P., D.B. and F.J.T.; resources, O.A.-P.; data curation, O.A.-P. and J.G.-R.; writing—original draft preparation, O.A.-P.; writing—review and editing, F.J.T. and J.G.-R.; visualization, O.A.-P. and D.B.; supervision, F.J.T. and J.G.-R. All authors have read and agreed to the published version of the manuscript.

**Funding:** This research received no external funding.

**Institutional Review Board Statement:** Not applicable.

**Informed Consent Statement:** Not applicable.

**Data Availability Statement:** All data and processing results are available upon request. They will be made available to anyone who submits a message to the corresponding author.

**Acknowledgments:** This was a self-funded investigation and applied to an area whose local community has no resources with which to implement these studies. Further thanks are due to the light seen at the end of a beer-night bet. The authors would also like to thank all those who will improve on the results obtained and who apply this methodology.

**Conflicts of Interest:** The authors declare no conflict of interest.

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
