# Peer review of "A Comparative Analysis for Defining the Sliding Surface and Internal Structure in an Active Landslide Using the HVSR Passive Geophysical Technique in Pujilí (Cotopaxi), Ecuador"

_land, doi:10.3390/land12050961_

Round 1

Reviewer 1 Report

Dear Authors,

The manuscript in its actual form looks more like a technical report, rather than a scientific research output. All the analysis has been well conducted and described, but in order to deserve the publication some advance, some new contribution to the research is expected.

What is the progress to the scientific research in the application of geophysical techniques to landslides investigations?

You should underline very well that eventual progress, otherwise the manuscript, in my opinion, is more a case study of the application of well known techniques, rather than a scientific paper.

Besides, the description of methods is too long and references about landslides could comprise more elements. Some minor points regarding maps are in the annotated version of the manuscript.

I encourage you in reducing the excessive length of methods section and implement a stronger discussion, highlighting the progress to scientific research; finally, conclusion must not be a summary of the manuscript.

Best regards

Author Response

Thank you

Reviewer 2 Report

The paper presents interesting applications of Horizontal to Vertical Spectral Ratio (HVSR) and Multichannel Analysis of Surface Waves (MASW) survey for defining rupture plane of landslide.  Accordingly, landslide rupture surfaces could be delineated based on the analyses.

In general, the paper is well written and the results are reported in detail. After reading the paper, the reviewer has the following comments:

1. Regarding “One of the first needs to study landslide mitigation actions is to define the rupture or failure surface. That can be an easy task when the mobilizing materials differ quite from the static ones. For example, when they are soft sediment sliding over a rocky substrate.”, please provide references to support the points of discussion.

2. Regarding “Traditional investigations consider performing drill holes and using instrumentation inside (like piezometers and inclinometers). These tests help define the sliding surface and provide information to make decisions about its mitigation. However, that kind of landslide study is usually expensive and complicated.”, please provide references to support the points of discussion.

3. Regarding “Some years ago, the application of passive ….”, please revise the sentence and be specific in time period.

4. It has been shown that in certain conditions, the HVSR method outcomes may vary seasonally, which must be considered when interpreting of the outcomes of this method. How did the authors tackle this issue?

5. Since the VSR method’s results depend on the properties of the soil above the rock. To some degree, noise can affect the results. Hence, the method may not be appropriate for all sites and may not always be successful or definitive. So, how to verify that the used methods are suitable for the geological conditions of the study area?

6. How consistent are the measured results obtained from the survey/test per site?

7. Can we use some quantitative measurement to discuss the economic benefits of the proposed methods?

8. Please elaborate the future extensions of the study in the concluding section.

Author Response

Thank you

Reviewer 3 Report

Dear authors,

here my suggestions to improve the manuscript:

General : I recommend an english review by an expert.

When you present the geophysical results - please, present the data, .. see eg seismic refraction interpretation section .. based on what? .. where is the tomography section obtained by inversion... also show at least in the annex some seismic signal recording.

Figure 1 presents only part of the landslide.. and so is difficult to be compared with Figure 2.

yours

reviewer H

Author Response

Thank you

Reviewer 4 Report

Dear editor,

Thank you for giving me the opportunity to revise the manuscript entitled “A comparative analysis for defining the sliding surface and in-2 ternal structure in an active landslide using the HVSR passive 3 geophysical technique. Pujilí (Cotopaxi), Ecuador” by Olegario Alonso-Pandavenes and his/her colleagues that was submitted to “Land”.

The paper presents a comparative analysis of different geophysical techniques for defining the sliding surface and internal structure of an active landslide in Pujilí, Ecuador. The authors applied 60 Horizontal to Vertical Spectral Ratio (HVSR) points passive seismic technique complemented with an active seismic refraction profile and Multichannel Analysis of Surface Waves (MASW) survey. They obtained the value of the ground's fundamental frequency fo and amplification Ao from HVSR, and the shear wave velocity for surface materials from active seismic surveys. They used two methods to estimate the thickness of the sediments over a compact material: applying the Nakamura equation and inverting the ellipticity curve using constrained models of shear wave velocity distribution. They also analyzed the directivity of the microtremor HVSR signals and its relation with the internal structure of the sliding material. They found that all three analyses (Nakamura equation, ellipticity inversion, and directivity analysis) gave consistent results for delineating the landslide rupture surface and validating the procedure. They also found that directivity analysis was useful for defining areas of most significant instability and vulnerability index Kg. They concluded that applying a simple technique such as HVSR was valid for preliminary studies in landslides and helpful in defining its rupture plane.

In general, it is an interesting work with certain academic values. However, there are still some problems need to be revised before manuscript being accepted.

Comments 1:

The discussion section of this paper was well-written, but there are some aspects that can be enhanced, such as:

1. It can be more succinct and polished, avoiding redundancy of the previous content or details.

2. It can emphasize its own novelty and contribution, rather than just reporting the results.

3. It can offer more concrete recommendations or insights, rather than just suggesting some ambiguous directions.'"

Comments 2:

The conclusion section of this paper was written fairly clearly, but there are some areas that can be improved, such as:

1. It can be more concise and refined, avoiding repeating the previous content or details.

2. It can highlight its own innovation and contribution, rather than just summarizing the results.

3. It can provide more specific suggestions or opinions, rather than just proposing some vague directions.”

Author Response

Thank you

Round 2

Reviewer 1 Report

Dear Authors,

In my opinion your manuscript now deserves the publication on Land.

Good luck and best regards

Author Response

Thank you and best regards.

Reviewer 3 Report

Dear authors,

I think the manuscript can now be accepted .. while the English is still not satisfactory all over the manuscript .. see e.g. sentence :

'According to early studies that were carried out in the area [3331 - 3634], that is due to the presence of a zone of poorly consolidated material that slides over a more compact material. '

.. that were .. that is due .. .. that slides ..

So, I would recommend another English review.

Also, the suggested improvements of the data presentation are quite limited.

yours

Reviewer H

Author Response

Thank you and best regards.
